# Regulating ion affinity and dehydration of metal-organic framework sub-nanochannels for high-precision ion separation

Ri-Jian Mo[1], Shuang Chen[1], Li-Qiu Huang[1], Xin-Lei Ding[1], Saima Rafique[1], Xing-Hua Xia [1] ✉ & Zhong-Qiu Li [1] ✉

Membrane consisting of ordered sub-nanochannels has been pursued in ion separation technology to achieve applications including desalination, environment management, and energy conversion. However, high-precision ion separation has not yet been achieved owing to the lack of deep understanding of ion transport mechanism in confined environments. Biological ion channels can conduct ions with ultrahigh permeability and selectivity, which is inseparable from the important role of channel size and "ion-channel" interaction. Here, inspired by the biological systems, we report the high-precision separation of monovalent and divalent cations in functionalized metal-organic framework (MOF) membranes (UiO-66-$(X)_2$, X = $NH_2$, SH, OH and $OCH_3$). We find that the functional group (X) and size of the MOF sub-nanochannel synergistically regulate the ion binding affinity and dehydration process, which is the key in enlarging the transport activation energy difference between target and interference ions to improve the separation performance. The $K^+$/$Mg^{2+}$ selectivity of the UiO-66-$(OCH_3)_2$ membrane reaches as high as 1567.8. This work provides a gateway to the understanding of ion transport mechanism and development of high-precision ion separation membranes.

Membrane-based separation technology has been nowadays widely used in desalination, water purification, and industrial production due to its low energy consumption and good integratability[1,2]. However, it yet faces various problems such as the disposal of concentrated waste solutions to the environment and the loss of valuable elements (*e.g.*, lithium, sodium, potassium, gold, platinum)[3–5]. To tackle these issues, urgent action is needed to develop membranes that can achieve high-precision "ion-ion" separation. However, this is hindered by the lack of molecular-level control of membrane structures and deep understanding of ion separation mechanisms.

Biological ion channels serve as a valuable inspiration in the development of high-precision separation membranes. A prime example is the $K^+$ ion channel, which shares the size of dehydrated $K^+$ ion (2.66 Å) and has a directional arrangement of carbonyl groups located on the surface of its protein lumen[6,7]. This configuration

exhibits a $K^+$/$Na^+$ selectivity of up to $10^4$. Ion transport through ion channel has to overcome an established ion transport activation energy of ion dehydration and its interaction with the channel surface[8]. For $K^+$ ion channel, its cavity can perfectly hold a dehydrated $K^+$ ion and keep it stable via $K^+$-carbonyl coordination. Furthermore, the channel can arrange up to four $K^+$ ions in close proximity, striking a balance between "ion-ion" repulsion and "ion-channel" attraction and leading to a low transport activation energy[9,10]. On the contrary, $Na^+$ ion struggles to enter and transport through the channel due to its unfavorable size and binding affinity, which leads to a high transport activation energy. Therefore, a channel that matches the size of the target ion and has appropriate interaction with the target ion plays a decisive role in achieving efficient ion separation[11].

Inspired by the biological ion channels, a range of biomimetic artificial channels have been developed for ion separation[12–15].

[1]State Key Laboratory of Analytical Chemistry for Life Science, School of Chemistry and Chemical Engineering, Nanjing University, 210023 Nanjing, China.
✉e-mail: xhxia@nju.edu.cn; zhongqiuli@nju.edu.cn

However, their performance is yet far from satisfactory[16-18]. Polymer membranes usually exhibit high stability, while their performance is limited by the lack of precise control over their pore structures[19,20]. Membranes with 2-dimensional laminated nanochannels have shown elevated ion selectivity, however, they usually suffer from poor stability and low ion fluxes[21-25]. Metal-organic framework (MOF) materials, including UiO-66-(COOH)$_2$[26], HKUST-1[27], MIL-53[28], and ZIF-8[29,30], have recently arisen as a promising alternative for addressing these shortcomings. The surface group and size of the well-ordered MOF sub-nanochannels can be precisely regulated by alternation of ligands and post-synthetic modification, affording a flexible platform for customizing the membrane properties. In addition, the high porosity and large specific surface area of MOFs ensure a large ion flux, resulting in great promise of circumventing the "permeability-selectivity" trade-off to further improve the separation performance[31-34]. Despite all these advantages, a comprehensive understanding of the influence of surface group and pore size of MOF channel on ion separation is urgently needed to reveal the underlying mechanism.

In this work, we systematically study the ion transport mechanism in the functionalized UiO-66-(X)$_2$ (X = NH$_2$, SH, OH, and OCH$_3$ (OMe)) sub-nanochannels (Fig. 1). Experimental results and theoretical simulations together revealed that functional group and size of the MOF sub-nanochannel can synergistically regulate the ion binding and dehydration processes to tune the ion transport energy barriers, resulting in different ion fluxes. The key to improve the separation performance lies in increasing the difference in ion transport activation energy between the target ions and interfering ions. The K$^+$/Mg$^{2+}$ selectivity of the UiO-66-(OMe)$_2$ membrane reaches as high as 1567.8. This work serves as a fundamental to comprehend ion transport mechanisms and advance the development of high-precision ion separation membranes.

## Results and discussion
### Characterization of UiO-66-(X)$_2$ membranes
UiO-66 and UiO-66-(X)$_2$ (X = NH$_2$, SH, and OH) membranes were prepared by a solvothermal method. Polyethylene terephthalate (PET) membrane with arrayed nanochannels (thickness, 12 μm; pore diameter, 227 ± 16.7 nm) was used as the substrate for MOF preparation (Supplementary Fig.1). The scanning electron microscopy (SEM) images (Fig. 2a-d and Supplementary Figs. 2-5) show that UiO-66 and UiO-66-(X)$_2$ uniformly grow along the PET substrate, forming a symmetrical structure that features a dense and crack-free MOF layer on both top and bottom surfaces, as well as MOF nanowires within the nanochannels. The X-ray diffraction (XRD) patterns show characteristic

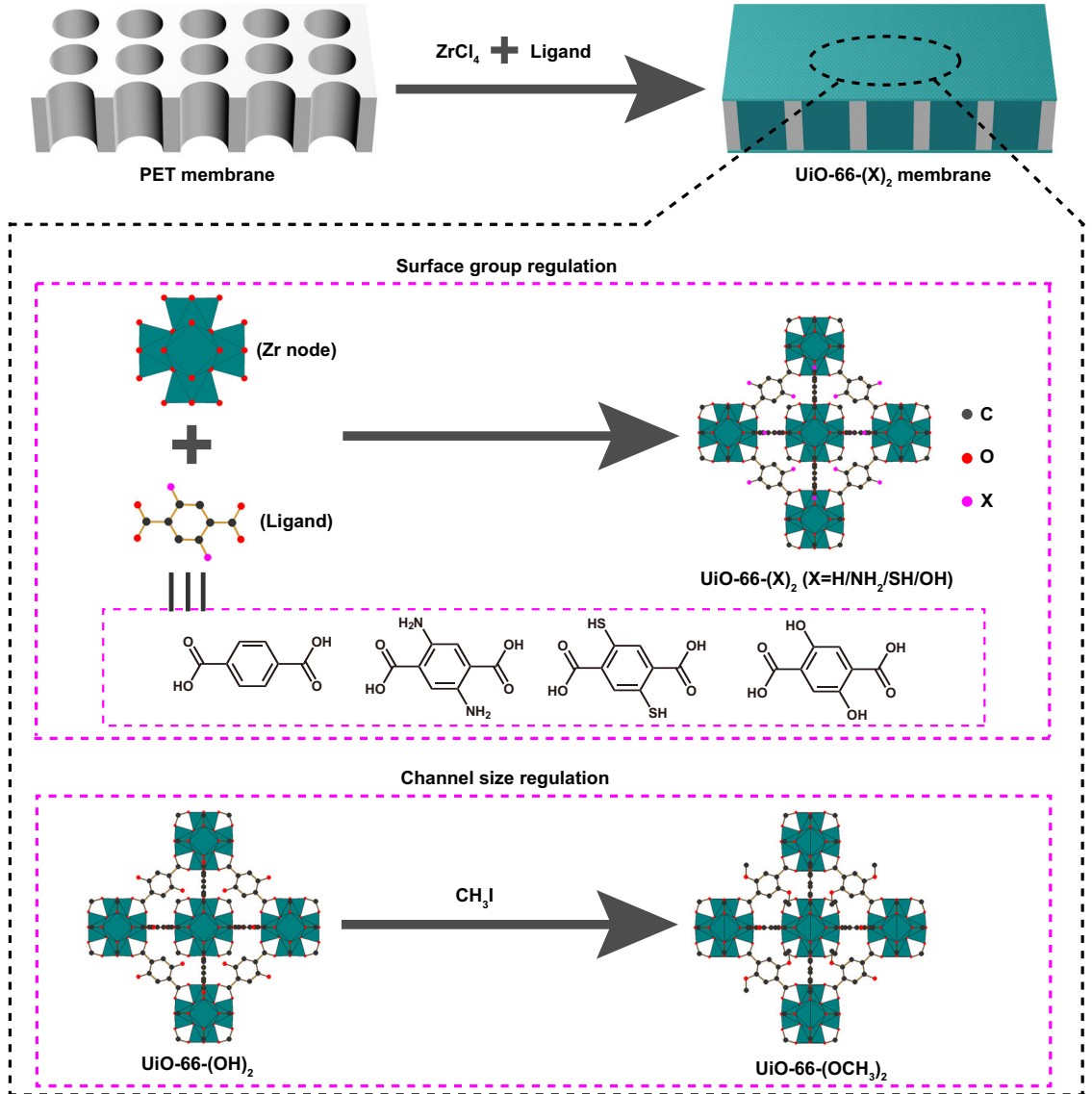

**Fig. 1 | Membrane fabrication.** Schematic illustration of the fabrication of UiO-66 and UiO-66-(X)$_2$ (X = NH$_2$, SH, OH, and OMe) membranes.

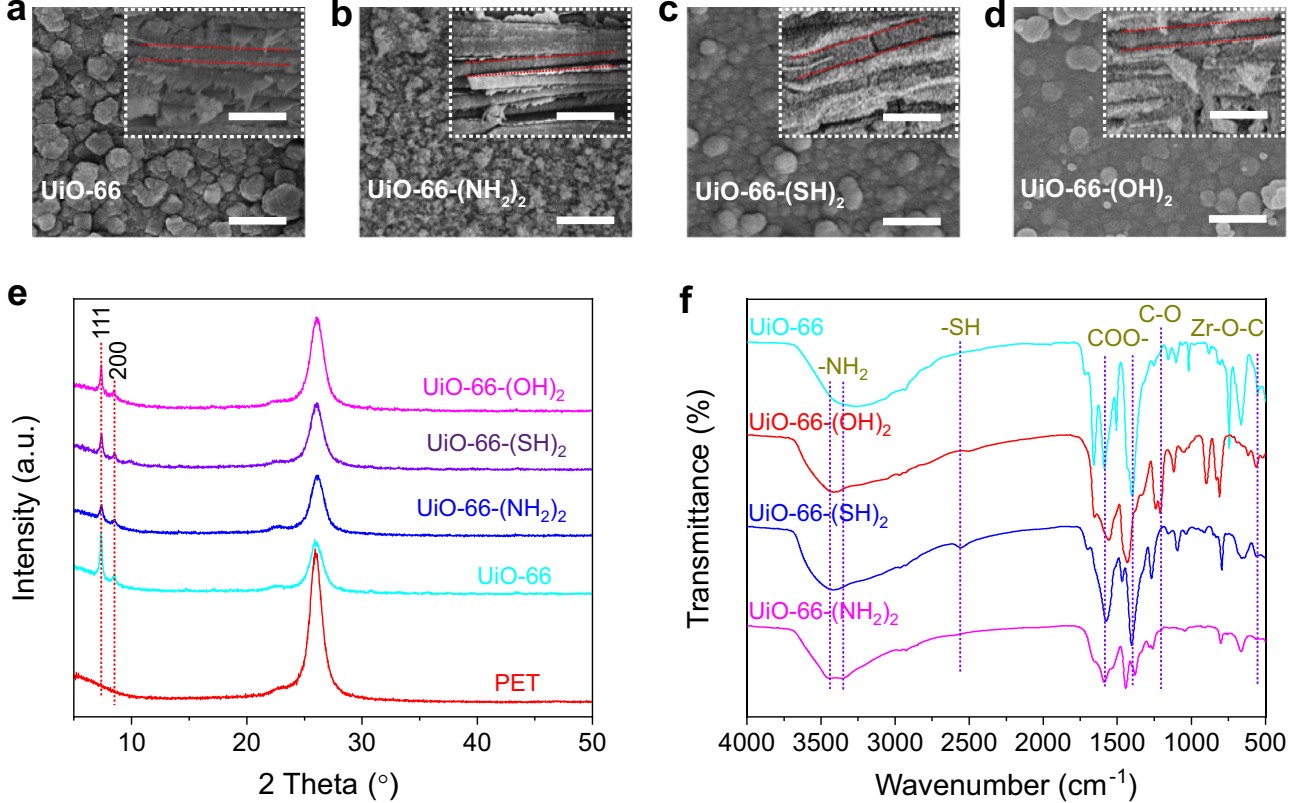

**Fig. 2 | Characterization of UiO-66-(X)$_2$ membranes. a–d** SEM images of UiO-66 and UiO-66-(X)$_2$ membranes. Inset shows the cross-section of the membranes. Scale bar is 1 μm in all the images. **e** XRD patterns of PET, UiO-66, and UiO-66-(X)$_2$ membranes. **f** FTIR spectra of UiO-66 and UiO-66-(X)$_2$. The peaks at 3366 and 3480 cm$^{-1}$ reflect the symmetric and asymmetric stretching vibration of -NH$_2$ bond in UiO-66-(NH$_2$)$_2$, respectively. The peak at 2572 cm$^{-1}$ reflects the stretching vibration of -SH bond in UiO-66-(SH)$_2$. The peak at 1238 cm$^{-1}$ reflects the stretching vibration of C-O on the benzene ring in UiO-66-(OH)$_2$. The peaks at 1395 and 1660 cm$^{-1}$ reflect the symmetric and asymmetric stretching vibrations of carboxylate groups of terephthalate acid, respectively. The peak at 557 cm$^{-1}$ reflects the stretching vibration of Zr-O-C.

diffraction peaks of UiO-66 at 7.4° and 8.5° (Fig. 2e and Supplementary Fig. 6), indicating that the crystal structure remains unchanged despite the introduction of -NH$_2$, -SH, or -OH groups[35,36]. The X-ray photoelectron spectroscopy (XPS) results reveal the characteristic peak belonging to Zr $3d$ for all the UiO-66-(X)$_2$ membranes, and peaks of N $1s$ and S $2p$ are observed for UiO-66-(NH$_2$)$_2$ and UiO-66-(SH)$_2$, respectively (Supplementary Fig. 7)[26]. The successful preparation of UiO-66-(X)$_2$ membranes was also supported by the Fourier transform infrared (FTIR) spectra (Fig. 2f). UiO-66 exhibits characteristic peaks at 557, 1395, and 1660 cm$^{-1}$[37]. For UiO-66-(OH)$_2$, an additional C-O stretching vibration peak of -OH on benzene is observed at 1238 cm$^{-1}$[38]. The symmetric stretching vibration and asymmetric stretching vibration peaks of -NH$_2$ are observed at 3366 and 3480 cm$^{-1}$ for UiO-66-(NH$_2$)$_2$[39]. A stretching vibration peak of -SH is observed at 2572 cm$^{-1}$ for UiO-66-(SH)$_2$[40]. The contact angles of UiO-66, UiO-66-(NH$_2$)$_2$, UiO-66-(SH)$_2$ and UiO-66-(OH)$_2$ membranes were measured to be 134.2°, 129.5°, 117.5° and 129.7°, respectively (Supplementary Fig. 8), indicating that all the membranes are slightly hydrophobic. Therefore, all the membranes were first wetted with ethanol to ensure the access of ions into the membrane in the following ion transport experiments. As revealed in Supplementary Fig. 9, the water contact angle of UiO-66-(OH)$_2$ membrane decreased from 127.9° to 103.4° after ethanol wetting, and it further decreased to 82.0° in 5 min. The porosity of UiO-66, UiO-66-(NH$_2$)$_2$, UiO-66-(SH)$_2$ and UiO-66-(OH)$_2$ was examined by nitrogen adsorption analysis at 77 K, and their Brunauer-Emmett-Teller surface areas were 1060.9, 587.9, 501.5 and 978.9 m$^2$ g$^{-1}$, respectively (Supplementary Fig. 10). The pore size distribution curve derived from the nitrogen adsorption isotherm indicates that the pore size of UiO-66-(X)$_2$ (X = NH$_2$, SH, and OH) is approximately 5.9 Å, exhibiting a similar size distribution.

## Separation mechanism: the role of surface groups

The interactions between ions and functional groups on the MOF channel can help grab and stabilize the dehydrated ions, but at the same time hinder their transport to the other side[11,41]. In order to study this effect, three surface groups (-NH$_2$, -SH and -OH) that have different coordination capabilities were chosen to regulate the ion binding affinity of the UiO-66-(X)$_2$ membranes. The zeta potential measurements show that, under the experimental conditions, UiO-66-(OH)$_2$ and UiO-66-(SH)$_2$ are negatively charged, while UiO-66-(NH$_2$)$_2$ is positively charged (Supplementary Fig. 11). The different charges together with the different polarizabilities of the three functional groups endow the MOFs with different binding energies toward ions, which is further demonstrated by DFT calculations. As shown in Supplementary Table 1, the electrostatic repulsion between the positively charged -NH$_2$ group and cations generates positive binding energies; while the electrostatic attraction between the negatively charged -SH/-OH group and cations generates negative binding energies. The ion binding affinity follows the order: UiO-66-(NH$_2$)$_2$ < UiO-66-(SH)$_2$ < UiO-66-(OH)$_2$. In addition, compared with monovalent cations (Li$^+$, Na$^+$ and K$^+$), divalent cations (Mg$^{2+}$ and Ca$^{2+}$) interact much more intensively with the functional groups due to their higher ionic charges.

The ion transport properties were studied by measuring the current-voltage (I-V) curves of UiO-66-(X)$_2$ membranes in 100 mM LiCl, NaCl, KCl, MgCl$_2$ and CaCl$_2$ aqueous solutions (pH 5.85) (Fig. 3a-c). For all the UiO-66-(X)$_2$ membranes, the ionic conductance derived from

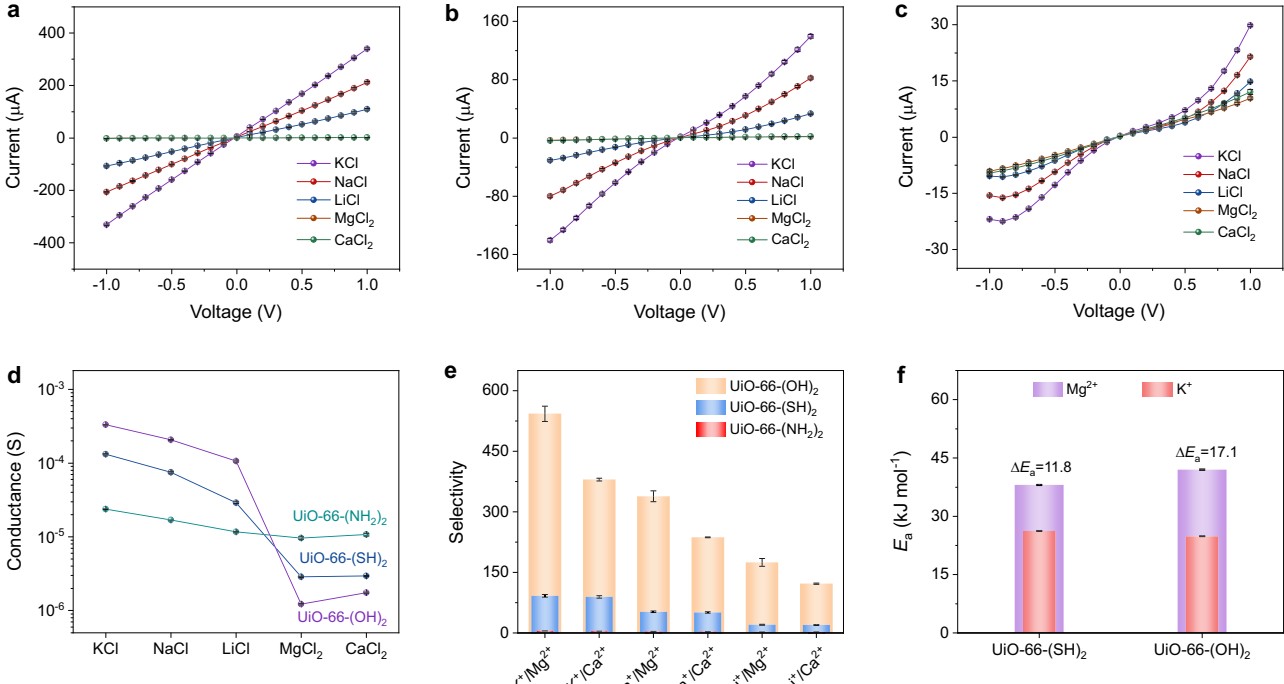

**Fig. 3 | Ion transport properties of UiO-66-(X)₂ membranes. a–c** I-V curves of UiO-66-(OH)₂, UiO-66-(SH)₂, and UiO-66-(NH₂)₂ membranes in different 100 mM electrolyte solutions. **d** Ionic conductance of UiO-66-(X)₂ membranes. **e** Ion selectivity of UiO-66-(X)₂ membranes. **f** Transport activation energies of K⁺ and Mg²⁺ ions in UiO-66-(SH)₂ and UiO-66-(OH)₂ membranes and corresponding activation energy differences ($\Delta E_a$) between K⁺ and Mg²⁺ ions. The error bars in all figures represent the standard deviation of three parallel tests.

the I-V curves, as shown in Fig. 2d, follows the order: KCl > NaCl > LiCl > CaCl₂ > MgCl₂. Additionally, when the functional group changes from -NH₂ to -SH and -OH, the conductance of monovalent cations through the membrane increases, while the conductance of divalent cations decreases, which leads to an increase in monovalent/divalent cation selectivity. This phenomenon is mainly attributed to the inferior binding affinities of the monovalent cations to the functional group as compared with the divalent cations. Therefore, for monovalent cations, a stronger binding energy can facilitate their entry into the MOF channel, leading to a larger ion flux; while for divalent cations, a stronger binding energy will hinder their transport within the MOF channel, leading to a lower ion flux.

The calculated monovalent/divalent cation selectivity of the UiO-66-(X)₂ membranes (Fig. 3e) follows the order: UiO-66-(NH₂)₂ ≪ UiO-66-(SH)₂ < UiO-66-(OH)₂ membrane. Taking K⁺/Mg²⁺ selectivity as an example, the values are 4.9, 92.0 and 542.9 for UiO-66-(NH₂)₂, UiO-66-(SH)₂, and UiO-66-(OH)₂ membranes, respectively. The separation performance of a bare PET and a UiO-66 membrane does not show any appreciable cation selectivity (Supplementary Figs. 12 and 13). It is noted that the conductance of monovalent cation in the UiO-66-(OH)₂ membrane is nearly equivalent to that in the bare PET membrane (Supplementary Fig. 14), indicating its unimpeded transmembrane transport.

The ion transport activation energies ($E_a$) through the UiO-66-(SH)₂ and UiO-66-(OH)₂ membranes would help to gain further understanding of the ion transport mechanism. It is observed that as the functional group changes from -SH to -OH, the transport activation energy of K⁺ decreases, while the one for Mg²⁺ increases, which ultimately raises the activation energy difference ($\Delta E_a$) between K⁺ and Mg²⁺ ions from 11.8 to 17.1 kJ mol⁻¹ (Fig. 3f and Supplementary Fig. 15a, b). Due to the exponential correlation between activation energy and ion flux, a significantly larger discrepancy in flux between K⁺ and Mg²⁺ ions is thus achieved. In addition, the $E_a$ values of K⁺ and Mg²⁺ ions through the UiO-66 and UiO-66-(NH₂)₂ membranes were also

measured (Supplementary Fig. 15c, d). The $\Delta E_a$ between K⁺ and Mg²⁺ ions follows the order: UiO-66-(NH₂)₂ < UiO-66 < UiO-66-(SH)₂ < UiO-66-(OH)₂, and the ion selectivity exponentially increases with the increase of $\Delta E_a$ (Supplementary Fig. 16). These results demonstrate that, through carefully regulating the functional group, the ion binding affinity can be effectively modulated to tune the ion transport activation energy. The greater the $\Delta E_a$ between the target and interference ions is, the larger the ion selectivity can be achieved.

Solution pH is an additional factor tuning the ion selectivity of UiO-66-(OH)₂ membrane via protonation and deprotonation. As shown in Supplementary Fig. 11, UiO-66-(OH)₂ exhibits an isoelectric point of approximately 3.8. With increasing the solution pH above the isoelectric point, -OH groups in MOF structure will become deprotonated, generating a strong binding affinity of cations to the channel. When the pH increases from 3 to 9, the conductance of K⁺ and Mg²⁺ ions shows a completely opposite trend: the conductance for K⁺ ion increases from 0.11 mS to 0.25 mS, while the one for Mg²⁺ ion shows a decrease from 2.4 μS to 0.59 μS (Supplementary Fig. 17). The K⁺/Mg²⁺ selectivity is thus increased from 86.9 to 857.9. This result further confirms that a stronger binding energy can facilitate the transport of monovalent cations but hinder the transport of divalent cations through the MOF sub-nanochannels.

The electrolyte concentration exhibits a great influence on ion separation performance. For K⁺ ion, as the concentration decreases, the conductance first linearly decreases, and then deviates from the bulk value when the concentration is lower than 10⁻⁴ M (Supplementary Fig. 18a, b). While for Mg²⁺ ion, the conductance keeps almost constant when the concentration is lower than 1 M, and gradually increases when the concentration is larger than 1 M (Supplementary Fig. 18c, d). These results indicate that, in the concentration range of 10⁻⁴ to 1 M, the transport of monovalent cations is controlled by the bulk solution, while the transport of divalent cations is determined by the surface property of the MOF sub-nanochannel[42–44]. Therefore, the monovalent/divalent cation selectivity of UiO-66-(OH)₂ membrane increases

with increasing the electrolyte concentration (Supplementary Fig. 18e), reaching a K$^+$/Mg$^{2+}$ selectivity as high as 1051.3 at the concentration of 1 M. With the electrolyte concentration exceeding 1 M, the K$^+$/Mg$^{2+}$ selectivity reaches a plateau because the transport of Mg$^{2+}$ ions in the high-concentration region gradually deviates from surface control (Supplementary Fig. 18e).

The ion selectivity is also determined by the content of -OH groups within the MOF structure. UiO-66-(H$_x$+OH$_{1-x}$)$_2$ membranes with different contents of -OH groups were synthesized by using ligands containing terephthalic acid and 2,5-dihydroxyterephthalic acid that mixed with different ratios. Results show that as the proportion of 2,5-dihydroxyterephthalic acid increases, the ion selectivity of the MOF membrane increases, reaching the highest value with pure 2,5-dihydroxyterephthalic acid as the ligand (Supplementary Fig. 19). The increase in the content of -OH group can offer more binding sites, which is beneficial to improve the ion selectivity. It is found that the K$^+$/Mg$^{2+}$ ion selectivity of the membrane remains basically unchanged within 120 h (Supplementary Fig. 20), and the integrity and crystallinity of the membrane were well maintained after the stability test (Supplementary Fig. 21), which demonstrates the high stability of the membrane.

## Separation mechanism: the role of channel size

With a size comparable to the hydrated ions (Supplementary Table 2), the MOF sub-nanochannel might cause ion dehydration and thus hinder the transport of ions with large size and high hydration energy[45]. The channel size of the UiO-66-(OH)$_2$ was regulated by attaching a methyl group to −OH group using the Williamson ether method (Fig. 1), forming UiO-66-(OMe)$_2$. The morphology (Supplementary Fig. 22) and the crystal structure (Supplementary Fig. 23a, b) of the UiO-66-(OMe)$_2$ membrane remain the same as those of the UiO-66-(OH)$_2$ membrane, and the water contact angle of UiO-66-(OMe)$_2$ membrane is also comparable to that of the UiO-66-(OH)$_2$ membrane, indicating an unvaried wettability (Supplementary Fig. 23c). The successful incorporation of the -OMe group into the MOF channel was confirmed by the appearance of a new C-H stretching vibration of the

-OMe group at approximately 2952 cm$^{-1}$ (Supplementary Fig. 24) in the FTIR spectrum and a characteristic peak of the -OMe group at 52.9 ppm in the $^{13}$C solid-state NMR spectrum (Supplementary Fig. 25). The porosity of UiO-66-(OMe)$_2$ was examined by nitrogen adsorption analysis at 77 K, and the Brunauer-Emmett-Teller surface area was 583.9 m$^2$ g$^{-1}$ (Supplementary Fig. 26). The pore size distribution curve derived from the nitrogen adsorption isotherm indicates that the pore size of UiO-66-(OMe)$_2$ is approximately 5.0 Å, which is slightly smaller as compared to that of UiO-66-(OH)$_2$.

The ion transport properties of the UiO-66-(OMe)$_2$ membrane were then investigated (Supplementary Fig. 27). When the functional group changes from -OH to -OMe, the conductance of monovalent cations remains almost the same; while the conductance of divalent cations prominently decreases (Fig. 4a). This indicates that, with decreasing the channel size, the transport of monovalent cations is not affected; while the transport of divalent cations is severely hindered. The K$^+$/Mg$^{2+}$ selectivity of the UiO-66-(OMe)$_2$ membrane is calculated as 1567.8, which is about 3 times larger than that of the UiO-66-(OH)$_2$ membrane (Fig. 4b). In addition, the ion selectivity of the UiO-66-(OMe)$_2$ membrane increases with increasing the electrolyte concentration, which is similar to that of the UiO-66-(OH)$_2$ membrane. The K$^+$/Mg$^{2+}$ selectivity of the UiO-66-(OMe)$_2$ membrane reaches as high as 3253.3 at the concentration of 1 M (Supplementary Fig. 28). It is also revealed that the K$^+$/Mg$^{2+}$ selectivity and the structure of the membrane can be well maintained within 120 h (Supplementary Figs. 29 and 30).

The ion transport in a decreased channel size requires the removal of more water molecules from its hydrated configuration, leading to an increased ion transport activation energy. As shown in Fig. 4c, d and Supplementary Fig. 31, when the functional group is changed from -OH to -OMe, the transport activation energy for K$^+$ ion only slightly increases due to its small size and low hydration energy; while for Mg$^{2+}$ ion that possesses larger size and higher hydration energy, its transport activation energy obviously increases. These changes in transport activation energy leads to an increased $\Delta E_a$ between K$^+$ and Mg$^{2+}$ ions from 17.1 to 22.2 kJ mol$^{-1}$, resulting in an increased ion selectivity.

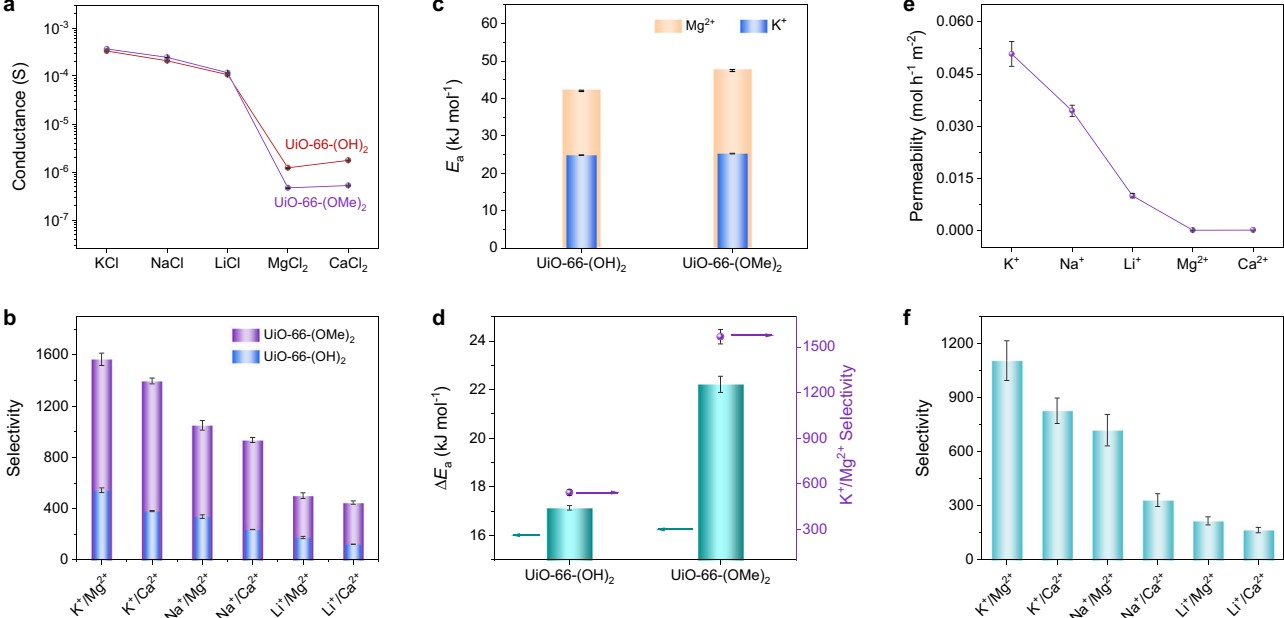

**Fig. 4 | Ion transport properties of UiO-66-(OMe)$_2$ membrane. a** Ionic conductance of UiO-66-(OH)$_2$ and UiO-66-(OMe)$_2$ membranes in different 100 mM electrolyte solutions. **b** Ion selectivity of UiO-66-(OH)$_2$ and UiO-66-(OMe)$_2$ membranes. **c** Transport activation energies of K$^+$ and Mg$^{2+}$ ions in UiO-66-(OH)$_2$ and UiO-66-(OMe)$_2$ membranes. **d** Transport activation energy differences ($\Delta E_a$) between K$^+$ and Mg$^{2+}$ ions and their relationship with K$^+$/Mg$^{2+}$ selectivity. **e, f** Ion permeability and permselectivity of the UiO-66-(OMe)$_2$ membrane. The error bars in all figures represent the standard deviation of three parallel tests.

To evaluate the practical application of the MOF membrane, the ion permselectivity of the UiO-66-(OMe)$_2$ membrane in a solution containing five different cations (0.5 M KCl, NaCl, LiCl, MgCl$_2$, and CaCl$_2$) was measured (Supplementary Fig. 32). The divalent cations exhibit much lower permeabilities as compared to the monovalent cations (Fig. 4e). The permselectivity of K$^+$/Mg$^{2+}$ was measured as 1048.4 (Fig. 4f), which represents the highest value over the reported thus far in the field of monovalent/divalent cation separation using artificial membranes (Supplementary Table 3).

Molecular dynamics (MD) simulation was conducted to gain further understanding of the mechanism for ion transport through the UiO-66-(OH)$_2$ and UiO-66-(OMe)$_2$ membranes (Supplementary Figs. 33 and 34). For UiO-66 type MOF, the ion transport pathway within the membrane is composed of a continuous repetition of "Cavity 1-Window-Cavity 2" (Fig. 5a). The potential of mean force (PMF) profiles of K$^+$ and Mg$^{2+}$ ions transported along the MOF channel were calculated to reveal the energy variation experienced by the ions during their migration (Fig. 5b, c). As summarized in Fig. 5d, when the functional group changes from -OH to -OMe, the transport energy barriers that K$^+$ and Mg$^{2+}$ ions need to overcome increase from 165.8 to 238.5 kJ mol$^{-1}$ and from 212.8 to 380.3 kJ mol$^{-1}$, respectively. The difference in transport energy barrier of K$^+$ and Mg$^{2+}$ ions thus increases from 47.0 to 141.8 kJ mol$^{-1}$. These results are consistent with those obtained from the activation energy experiments (Fig. 4e). In order to rule out the influence of "ion-channel" interaction, the binding energies of K$^+$ and Mg$^{2+}$ ions to the -OMe group were calculated, and they both show a decrease as compared with those of K$^+$ and Mg$^{2+}$ ions to the -OH group (Supplementary Table 1). The decrease in binding energy is supposed to be detrimental to ion separation. However, an improved ion separation performance is observed in our experiments (Fig. 4b), which demonstrates that the channel size should play a dominant impact in the ion separation.

The hydration states of the ions at different locations along the pathway were then extracted to further explore the ion transport mechanism at the molecular level. The coordination numbers of water molecules in the first hydration layer of K$^+$ and Mg$^{2+}$ ions were obtained by PMF calculations (Fig. 5e-j). It is revealed that the transport of ions experiences dehydration during passing through the window, and then rehydration when entering into the cavity again. This repetitive "dehydration-rehydration" process of the transported ions coincides with the results from radial distribution function (RDF) calculations (Supplementary Fig. 35). When the functional group changes from -OH to -OMe, the ion-water coordination numbers of K$^+$ and Mg$^{2+}$ ions at the window region decrease from 3.8 to 2.4 and from 5 to 3, respectively. In order to compare the energy consumption of K$^+$ and Mg$^{2+}$ ions during their dehydrations, the RDF profiles of K$^+$-water and Mg$^{2+}$-water at the window region are further analyzed. There is a clear boundary between the first and second hydration layers of Mg$^{2+}$ ion (Supplementary Fig. 35a, d), indicating that the water molecules in the first hydration layer are tightly bonded to the ion core; while for K$^+$ ion, the boundary is relatively blurred, indicating that the water molecules in the first hydration layer are loosely bonded to the ion core and can quickly exchange with the water molecules in the second hydration layer[46]. This demonstrates that Mg$^{2+}$ ion is more difficult to dehydrate as compared with K$^+$ ion, which is in accordance with the hydration energy of K$^+$ and Mg$^{2+}$ ions (Supplementary Table 2). Therefore, with reducing the channel size, the energy barrier increment of Mg$^{2+}$ ion is much larger than that of K$^+$ ion, thus an improved ion separation performance is observed (Supplementary Fig. 36). Similar results for K$^+$ and Mg$^{2+}$ ions entering the UiO-66-(OH)$_2$ and UiO-66-(OMe)$_2$ membranes from bulk solution are obtained (Supplementary Fig. 37). These results demonstrate that the size of MOF channel can effectively tune the ion dehydration process and improve the ion separation performance.

In conclusion, we have demonstrated that high-precision separation of monovalent and divalent cations with selectivity of several thousands can be achieved using MOF membranes by molecularly tailoring the functional group and pore size that regulate the ion affinity and ion dehydration, respectively. In addition, we find an exponential relationship between ion selectivity and "target-interference" ion activation energy differences, which offers a strategy to design high-performance separation membranes. To further reveal the ion separation mechanism, in-situ spectroscopic and electron microscopic techniques are expected to be developed to investigate the hydration state of transported ions and evaluate the interactions between ions and the channel surface. The present work sheds a light on the membrane separation mechanism and provides a strategy to design high-performance MOFs-based separation membranes for environmental preservation and energy production.

## Methods

### Material and chemicals

PET membranes (12 μm in thickness; $3 \times 10^8$ pores cm$^{-2}$) were provided by Wuwei Kejin Xinfa Technology Co., Ltd. Zirconium chloride (ZrCl$_4$) and N,N-dimethylformamide (DMF) were purchased from Shanghai Aladdin Biochemical Technology Co., Ltd. Terephthalic acid, 2,5-dihydroxyterephthalic acid and anhydrous potassium carbonate were provided by Shanghai Macklin Biochemical Technology Co., Ltd. 2,5-Diaminoterephthalic acid and 2,5-dimercaptoterephthalic acid were purchased from Jilin Zhongke Technology Co., Ltd. and Shanghai Bide Pharmaceutical Technology Co., Ltd., respectively. Methyl iodide (MeI) was obtained from Shanghai Titan Technology Co., Ltd. Ethanol, methanol, acetic acid, potassium chloride, sodium chloride, lithium chloride, magnesium chloride and calcium chloride were from China Sinopharm Chemical Reagent Co., Ltd. All the chemicals were used without further purification. Deionized water (>18 MΩ•cm) was produced by Milli-Q Water System (Millipore, Nanjing, China).

### Preparation of UiO-66-(OH)$_2$ membrane

A PET membrane (diameter, 12 mm) was placed vertically into a 25 mL reaction kettle. Then, 0.3 mmol ZrCl$_4$ and 0.3 mmol 2,5-dihydroxy terephthalic acid were dissolved in a mixture of 10 mL DMF and 0.5 mL acetic acid by ultrasound. After dissolution, the mixture was poured into a reaction kettle and reacted at 120 °C for 48 h. The resulting UiO-66-(OH)$_2$ membrane was washed with ethanol and stored in ethanol. UiO-66, UiO-66-(NH$_2$)$_2$ and UiO-66-(SH)$_2$ membrane were prepared with the similar method as UiO-66-(OH)$_2$ membrane.

### Preparation of MOF powders

After membrane synthesis, UiO-66, UiO-66-(NH$_2$)$_2$, UiO-66-(SH)$_2$ and UiO-66-(OH)$_2$ powders were collected from the mother liquor in the above autoclave. The powders were washed with DMF and methanol centrifugation (12000 rpm, 20 min), respectively.

### Preparation of UiO-66-(H$_x$ + OH$_{1-x}$)$_2$ membrane

The UiO-66-(H$_x$ + OH$_{1-x}$)$_2$ membrane was prepared by a solvothermal method using mixed organic ligands. Specifically, a PET membrane was placed vertically into a 25 mL reaction kettle. 0.3 mmol of ZrCl$_4$, $n$ mmol ($n$ < 0.3) of terephthalic acid, and (0.3-$n$) mmol of 2,5-dihydroxy terephthalic acid were dissolved in 10 mL DMF and 0.5 mL acetic acid by ultrasound. The mixture was poured into the reaction kettle and reacted at 120 °C for 48 h. The obtained membrane was washed with ethanol, stored in ethanol, and labeled as the UiO-66-(H$_x$ + OH$_{1-x}$)$_2$ membrane.

### Preparation of UiO-66-(OMe)$_2$ membrane and UiO-66-(OMe)$_2$ powder

UiO-66-(OMe)$_2$ membrane was prepared by a post-modification method. A 1.2 g anhydrous K$_2$CO$_3$ and 20 mL anhydrous DMF were added in a 50 mL round-bottom flask, and then 60 μL MeI solution was

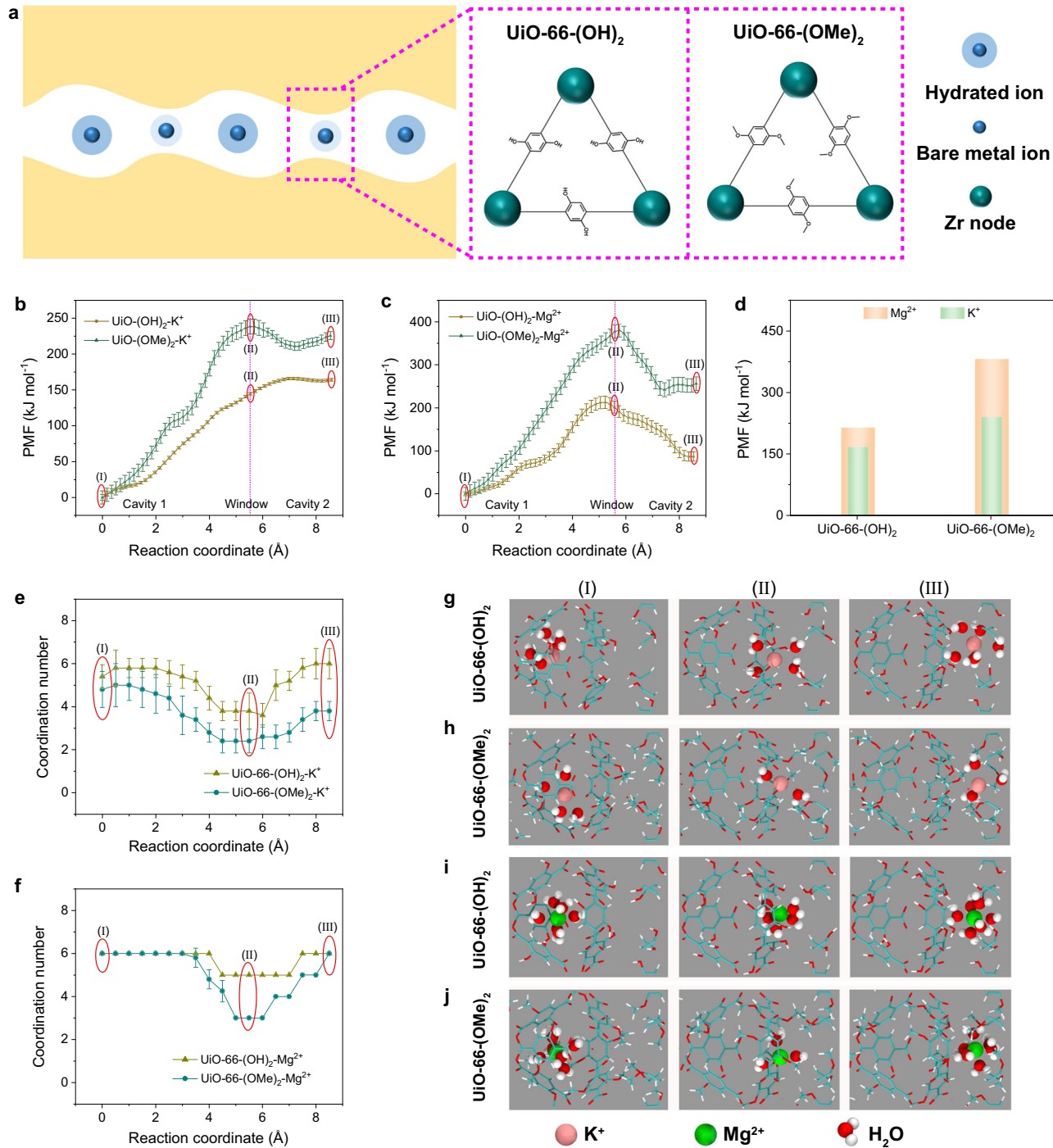

**Fig. 5 | MD simulation of ion transport in UiO-66-(OH)$_2$ and UiO-66-(OMe)$_2$ membranes.** **a** Ion transport pathway in UiO-66-type membrane. **b** Variation of PMF for K$^+$ during its transport. **c** Variation of PMF for Mg$^{2+}$ during its transport. The pink dashed line in **b** and **c** stands for the location of window. **d** Energy barriers for K$^+$ and Mg$^{2+}$ ions to transport. **e**, **f** The ion-water coordination numbers of K$^+$ and Mg$^{2+}$ ions during their transport through UiO-66-(OH)$_2$ and UiO-66-(OMe)$_2$ membranes. Locations of (I), (II), and (III) in **b**, **c**, **e** and **f** (marked by red circles) represent "Cavity 1", "Window", and "Cavity 2" regions, respectively. **g**, **h** Molecular snapshots of K$^+$ ion at locations of (I), (II), and (III) in UiO-66-(OH)$_2$ and UiO-66-(OMe)$_2$ membranes. **i**, **j** Molecular snapshots of Mg$^{2+}$ ion at locations of (I), (II), and (III) in UiO-66-(OH)$_2$ and UiO-66-(OMe)$_2$ membranes. The error bars in **b** and **c** represent the standard deviation of the data collected in the final 6 ns of the simulation performed at each location. The error bars in **e** and **f** represent the standard deviation of five parallel tests.

added with a vigorous stirring. The mixture was allowed to stand for 5 min. Then, UiO-66-(OH)$_2$ membrane was added in the flask, and the whole set-up was placed in a water bath at 87 °C for 10 h. After the reaction, the membrane was washed with ethanol and stored in ethanol. The preparation of UiO-66-(OMe)$_2$ powder was similar to that of the membrane, except that 20 mg UiO-66-(OH)$_2$ powder was added into the reaction flask.

## Characterizations

XRD patterns of the samples in a 2θ range of 3-80° were recorded using a D8 ADVANCE diffractometer (Brock, Germany) under Cu-Kα radiation (40 kV and 20 mA) at room temperature and at a scanning speed of 0.2 or 0.4 sec/step. The surface and cross-section morphologies of the membrane samples were examined using a field emission scanning electron microscope (JEOL JSM-7800F, Japan) at 5 kV. All the samples

were coated with gold before examination. FTIR spectra of samples were recorded in the range of 500 to 4000 cm$^{-1}$ using KBr particles on a Fourier transform infrared spectrometer (ThermoFisher Nicolet 6700, USA). A Zeta analyzer (Nano-Z, UK) was used to analyze the Zeta potentials of MOF powders in aqueous solutions. The concentrations of MOF samples were 0.05 mg/mL. Solid-state NMR spectra were acquired on a Bruker NMR spectrometer (AVANCE AV III 400WB). The wettabilities of MOF membranes were analyzed using a Contact angle meter (DataPhysics OCA30, Germany). XPS spectra were recorded using a PHI 5000 VersaProbe-III electron spectroscopy (Japan). Nitrogen sorption isotherms were measured using a Micromeritics ASAP 2460 Multi-station automatic specific surface and aperture analyzer. A liquid nitrogen bath was used to maintain a temperature of 77 K for each measurement. High-purity (99.999%) nitrogen was used throughout the adsorption experiments. The surface areas and pore sizes of MOF samples can be determined by calculating nitrogen sorption isotherms.

## Ion current measurement

The ion transport properties of the as-prepared MOF membranes were investigated using linear sweep voltammetry (LSV). The MOF membranes were first clapped between two PDMS gaskets, and then placed between the two electrolytic half cells. All the membranes were first wetted with ethanol, which ensures the access of ions into the membrane in the following ion transport experiment. Each cell was filled with electrolyte solution with the same volumes, and a pair of Ag/AgCl reference electrodes was used to apply transmembrane voltage. The testing area of each membrane was approximately 3.14 mm$^2$. Keithley 2450 instrument was used to acquire the I-V curves in the range from -1 to +1 V at a scan rate of 50 mV s$^{-1}$. Each test was repeated at least three times to obtain the average current value at different voltages. The pH of the electrolyte solution was adjusted by adding 1 M NaOH or HCl solutions. Unless otherwise stated, the pH of the as-prepared electrolyte solution was 5.85.

## Ion selectivity calculation based on ion conductance

Due to the different valence states carried by monovalent and divalent cations, their contributions to the ionic current will be different. For quantitative comparison, the calculation of ion selectivity needs to take the valence state into account. Therefore, the metal ion selectivity ($S$) is calculated as follows:

$$S = \frac{G_M}{G_{M^*}} \times \frac{z_1}{z_2} \tag{1}$$

where $G_M$ and $G_{M^*}$ are the conductivity of the metal chlorides M and M* through the membrane, respectively; $z_1$ and $z_2$ are the valence states of the metal ions in M* and M, respectively.

## Ion transport activation energy calculation

The ion transport activation energy ($E_a$) can be calculated according to the Arrhenius-type equation[8,47]:

$$\ln(GT) = \ln(\beta) - \left(\frac{E_a}{R} \cdot \frac{1}{T}\right) \tag{2}$$

where $G$ is the conductance of the ion across the membrane, $T$ is the absolute temperature, $\beta$ is the pre-exponential factor, $R$ is the gas constant.

The experiment process is as follows: first, the membrane was sandwiched between two temperature-controlled half cells; second, the I-V curve was recorded using an electrochemical workstation (CHI 660E) to obtain the corresponding ion conductance at each temperature when the set-up reached a steady state; third, the activation energy was calculated according to the formula by plugging the

obtained conductance values. The temperature range used in this work was from 22 to 37 °C.

## Ion permeation experiment

The ion permeation experiment was carried out in a homemade H-type electrolytic cell (Supplementary Fig. S32). The UiO-66-(OMe)$_2$ membrane was mounted between the two electrolytic half cells. The test area was approximately 3.14 mm$^2$. A 35 mL mixed solution, which contains LiCl, NaCl, KCl, MgCl$_2$, and CaCl$_2$ (0.5 M for each electrolyte), was added into one of the half cells as the feed solution, and 35 mL of deionized water was filled into the other half cell as the permeation solution. Two Pt electrodes were placed in the electrolytic cells. The ion permeation experiment was performed by applying a constant trans-membrane voltage of 1 V for 6 h with an electrochemical workstation (CHI 660E). In order to avoid the concentration polarization effect, both the feed and the permeation chambers were magnetically stirred. At the end of the experiment, the concentrations of five different cations in the permeation solution were measured by inductively coupled plasma optical emission spectrometry (ICP-OES, iCAP 7000, USA).

## Density functional theory calculation

Periodic plane wave DFT calculations were utilized to determine the adsorption energies of K$^+$, Na$^+$, Li$^+$, Mg$^{2+}$, and Ca$^{2+}$ cations on the UiO-66, UiO-66-(NH$_2$)$_2$, UiO-66-(SH)$_2$ and UiO-66-(OH)$_2$. These calculations were carried out using the Vienna Ab-initio Simulation Package (VASPSol) with the projector augmented plane wave (PAW) pseudo potential basis set and generalized gradient approximation (GGA) functional according to the Perdew-Burke-Ernzerhof (PBE) parametrization[48-51]. For all systems, a kinetic energy cutoff of 520 eV was utilized for the plane wave expansion. To ensure high accuracy, the brillouin zone integration was sampled using the Monkhorst-Pack scheme with a 4 × 4 × 4 k-grid mesh[52]. The effect of cavitation, electrostatics, and dispersion of the solvent of water was depicted using the implicit solvation model that implemented in VASPSol. Finally, the adsorption energies were obtained using the following equation:

$$E_{ab} = E_{MOF-M^+} - E_{MOF} - E_{M^+} \tag{3}$$

where $E_{MOF-M^+}$ is the total energy of the MOF-M$^+$ system; $E_{MOF}$ and $E_{M^+}$ are the energies of the MOF structure and metal ions in solvent, respectively.

## Molecular dynamics simulation

MD simulations performed in this study utilized the NVT ensemble and were conducted using the GROMACS 4.6.7 package[53]. The force field parameters for UiO-66 and ions were adopted from a previous report[26]. During the simulations, all heavy atoms in UiO-66 were kept fixed, while periodic boundary conditions were applied along all three directions. To maintain a temperature of 298 K, the Nosé-Hoover thermostat was employed[54]. The long-range electrostatic interactions were calculated using the particle mesh Ewald (PME) summation method, with a cutoff of 1.3 nm for distinguishing between direct and reciprocal space summations[55]. The cutoff distance was also set to 1.3 nm for the van der Waals interactions. The parameters for the Lennard-Jones potential, governing the interactions among nonbonded atoms, were determined using the Lorentz-Berthelot combination rule[56]. Finally, a time step of 1 fs was used in the simulations.

The cellulose (110) systems consisted ~17000 atoms with a dimension of nearly 5.32 × 7.58 × 6.22 nm$^3$ for all MD simulations, which were constructed by a UiO-66-type (that is, UiO-66-(OH)$_2$ or UiO-66-(OMe)$_2$) crystal of 5.14 × 2.97 × 3.45 nm$^3$ and ~4330 TIP3P water molecules[57]. To calculate the free energy profiles (potentials of mean forces, PMFs) for ions entering the channel of UiO-66-type crystal, the umbrella sampling method was employed[58]. For each system, the

reaction pathway was divided into equally spaced windows of 0.01 nm. For each umbrella window, a 10 ns simulation was performed. To confine the ions motion along the z-axis, a harmonic force with a stiffness ranging from 600 ~ 20,000 kJ mol$^{-1}$ nm$^{-2}$ was applied. The initial 4 ns of each simulation run, known as the equilibration phase, were excluded from further analysis, while the final 6 ns of each simulation were employed to generate the final PMF profiles using the g_wham utility of GROMACS.

### Reporting summary

Further information on research design is available in the Nature Portfolio Reporting Summary linked to this article.

## Data availability

All data generated or analyzed during this study are available in this article and its Supplementary Information files. Source data are available from corresponding authors upon request.

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

## Acknowledgements

This work was supported by the National Key R&D Program of China (2017YFA0700500), the National Natural Science Foundation of China (22074061, 22204071), the Natural Science Foundation of the Jiangsu Province (BK20220770), and the Excellent Research Program of Nanjing University (ZYJH004).

## Author contributions

X.X. and Z.L. conceived ideas and supervised the project. R.M. and S.C. performed the experiments and the related characterization. Z.L. and R.M. performed the molecular dynamics simulations. R.M. and Z.L. analyzed the data and wrote the manuscript. X.X. and Z.L. revised the manuscript. L.H., X.D. and S.R. discussed the results and commented on the manuscript.

## Competing interests

The authors declare no competing interests.
