## [Peer Review File · Nature Communications]

REVIEWER COMMENTS

Reviewer #1 (Remarks to the Author):

This manuscript prepared a series of UiO-66-(X)₂ (X=NH₂, SH, OH, and OMe) MOF membranes by growing MOFs into/onto the PET nanochannel membranes. The UiO-66-(OMe)₂ membranes exhibited the best mono-/di-valent cation selectivity. MD simulation is employed to investigate the mechanism of ion transport through modified UiO-66 channels. This manuscript is well-written. However, it is unclear about the concept novelty of the work over previously published works of UiO-66-X MOF channels/membranes for cation and/or anion sieving. Besides, more experimental pieces of evidence should be provided to support the arguments in the manuscript. Some comments are given to be considered by the authors to improve the quality of this manuscript.

1. This work used PET membrane as a substrate for MOF preparation. This manuscript provides the thickness (10 μm) and pore diameter (220 nm) of the PET substrate. What is the channel density of the PET membranes? Is it a commercial PET membrane or prepared using a chemical etching method? If the 220 nm pore diameter is a statistical result, the author should provide the distribution and error bar of this data. It would be great if the effects of the pore size and pore density of the substrate on the quality of the MOF membranes could be studied and analyzed in this work.
2. From Figure S1a, many pores coalesced together into larger pores. Does it affect the performance of MOF membranes? The authors should add more details of the PET membrane they used in this work. In addition, why does Figure S1b measure the thickness of the PET membrane as 12 μm, not 10 μm? Which data is correct?
3. The authors claimed that monovalent/divalent cation selectivity of the UiO-66-(X)₂ membranes (Figure 3e) follows the order: UiO-66-(NH₂)₂ << UiO-66-(SH)₂ < UiO-66-(OH)₂ membrane. It would be better to provide the pore size distribution of the four tested MOF crystals to help understand the relationship between MOF pore sizes and membrane selectivity. In addition, it would be better to explain whether the ion selectivity observed in this work mainly depends on the membrane quality other than the MOF pore sizes because -SH group is larger than -OH group, and UiO-66-(SH)₂ should have better ion selectivity than UiO-66-(OH)₂ pores.
4. What is the diameter of the prepared MOF membrane? And what is the size for ion permselective testing?
5. The contact angles of UiO-66-(X)₂ membranes were measured (Figure S7) and all the membranes were slightly hydrophobic. From the SEM images of the UiO-66-(X)₂ membranes (Figure 2), the surfaces of the membranes were fully covered by MOF crystals. Thus, do you mean the UiO-66-(X)₂ are all hydrophobic?
6. The difference in mono-/di-valent ion selectivity was attributed to the inferior binding affinities (Page 6). Why could a stronger binding energy result in different ion flux trends for monovalent cations and divalent cations? A much clearer description is required here.
7. On page 7, the authors said, 'the conductance of monovalent cation in the UiO-66-(OH)₂ membrane is nearly equivalent to that in the bare PET membrane'. Data and figures should be given to prove this.
8. The purple line in Figure S12 is meaningless. It's not a curve to exhibit the data from the same samples. The purple line in Figure 4d is meaningless as well.
9. From Figure S14, the mono-/di-valent cation selectivity of the membranes increases with the increase of electrolyte concentration. Will it keep increasing when the electrolyte concentration is higher than 1 M?
10. Figure S17 exhibited the stability of UiO-66-(OH)₂ membranes within 48 h. How did the authors do this experiment? Was the membrane kept working under an electric field for 48 hours or just kept in water? More details should be provided to clarify what the stability is about. In addition, 48 h is not enough to exhibit the stability of the membrane.
11. UiO-66-(OMe)₂ membrane was obtained by post modification of UiO-66-(OH)₂ membrane. Is it only modified on the surface of UiO-66-(OH)₂ or fully modified even the MOFs into the PET channel? How do the authors identify the modification ratio of UiO-66-(OH)₂ MOFs?
12. To evaluate the ion permselectivity of UiO-66-(OMe)₂ membrane, a mixed solution of 0.5 M KCl, NaCl, LiCl, MgCl₂, and CaCl₂ was prepared for testing. Why use this concentration in this experiment? In the other experiments, the authors used 0.1 M or 1 M for testing.
13. It would be better to provide the MOF membrane mechanical properties for demonstrating

their potential for separation application.

Reviewer #2 (Remarks to the Author):

In this work the authors prepared various functional metal organic framework (MOF) membranes (UiO-66-(X)₂, X=NH₂, SH, OH and OCH₃) and investigated the ionic transportation behavior through these membranes under electric field. The resulted membranes demonstrated good separation performance for the separation of monovalent and divalent cations due to the functional groups (X) and size of the MOF sub-nanochannel synergistically regulate the ion binding affinity and dehydration process. This may provide some clue for the development of high-precision ion separation membranes.

1. The authors claimed they were inspired by the biological systems and designed the resulted membrane. The relationship between them is not so close. It is better to delete it.
2. The synthesis process is simple. Both MOFs powders and crystals were formed on the surface of PET substrates, and in the solution. How to control the formation process only on the surface? How to guarantee the uniformity of MOFs on the surface and in the channels? Namely, why can MOFs crystals grow uniformly into the channels and on the surface? How about the effect of the channel size and surface property on the growth process?
3. The conductivity of pure bulk solution of the electrolytes should be given.
4. How to determine the distribution of the mixture linkers into the MOFs? doping or just a mixture MOFs?
5. From the SEM cross-section, it is hard to see the full filled nanowires. It is better to remove the PET substrate and carry out the SEM or TEM observation.
6. Why most of the I-V curves are not linear in the -1V to 1V?
7. It is better to determine the activation energy by verifying temperature. This is more directly.
8. In Figure S17, Why the conductance for both KCl and CaCl₂ increase with time but selectivity decrease with time. Some leaking exist?
9. Since the membrane are composed of nanocrystals, many crystal grain boundary should be exist. Please address the contribution of these boundary to the ionic transportation.
10. From the water contact angles, the surfaces of the membranes are very hydrophobic. How can water fill into the nanochannels under the conditions as authors claimed.
11. Is it possible to perform the membrane for pressure-driven ionic separation? Pressure-driven ionic separation is more facile.

Reviewer #3 (Remarks to the Author):

This manuscript titled in "Regulating Ion Affinity and Dehydration of Metal-Organic Framework Sub-nanochannels for High-Precision Ion Separation" by Xia et al. reports the UiO-66-(OCH₃)₂ membrane to achieve the high K⁺/Mg²⁺ selectivity as 1567.8. This work provides a new gateway to the understanding of ion transport mechanism and development of high-precision ion separation membranes. I would recommend the acceptance of the manuscript for a major revision after addressing the following issues.

1. Why do not the I-V curves in Figure 3c follow the Ohm's law?
2. "When the electrolyte concentration is increased from 10⁻⁴ to 1 M, the conductance of K⁺ ion linearly increases, while the conductance of Mg²⁺ ion keeps almost constant (Figure S14)." Why can it show excellent surface charge control at a high concentration of 1 M? Referring to other literatures, the electric double layer will become very thin at high concentrations.
3. Why can the transport of monovalent cations be controlled by the bulk solution? Why can not the transport of monovalent cations be controlled by the surface charge? Here needs more discussions.
4. The stability test time (48 h) is too short to explain high stability of the membrane.
5. Why use UiO-66-(OH)₂ as the membrane rather than UiO-66-(COOH)₂? Here needs more discussions.

RESPONSE TO REVIEWERS' COMMENTS

Reviewer #1

This manuscript prepared a series of UiO-66-(X)₂ (X=NH₂, SH, OH, and OMe) MOF membranes by growing MOFs into/onto the PET nanochannel membranes. The UiO-66-(OMe)₂ membranes exhibited the best mono-/di-valent cation selectivity. MD simulation is employed to investigate the mechanism of ion transport through modified UiO-66 channels. This manuscript is well-written. However, it is unclear about the concept novelty of the work over previously published works of UiO-66-X MOF channels/membranes for cation and/or anion sieving. Besides, more experimental pieces of evidence should be provided to support the arguments in the manuscript. Some comments are given to be considered by the authors to improve the quality of this manuscript.

Response: We thank this reviewer for the valuable and helpful comments that help enhance the quality of our manuscript.

We are sorry for the unclear description of the concept novelty of our work. As compared with the previous studies on UiO-66-X MOF membranes for ion sieving, our research places greater emphasis on understanding the ion transport mechanism at the sub-nanoscale level. We successfully establish a connection between several important factors, that is, the “functional group and size of the channel” can synergistically regulate the “ion binding and dehydration” processes to tune the “ion transport energy barriers”, resulting in different “ion fluxes”. Our findings demonstrate that the key to improve the separation performance lies in increasing the difference in ion transport activation energy between the target ions and interfering ions. We have revised our manuscript to clearly emphasize the novelty of our work. The modified sentences are as follows:

“In this work, we systematically study the ion transport mechanism in the functionalized UiO-66-(X)₂ (X=NH₂, SH, OH, and OCH₃) sub-nanochannels (Figure 1). Experimental results and theoretical simulations together revealed that functional group and size of the MOF sub-nanochannel can synergistically regulate the ion binding and dehydration

processes to tune the ion transport energy barriers, resulting in different ion fluxes. The key to improve the separation performance lies in increasing the difference in transport activation energy between the target ions and interfering ions. The K^+/Mg^{2+} selectivity of the $UiO-66-(OCH_3)_2$ membrane reaches as high as 1567.8. This work serves as a fundamental to comprehend ion transport mechanisms and advance the development of high-precision ion separation membranes.” (Page 3, revised manuscript)

In addition, according to the suggestions and comments, we have supplemented more experimental pieces of evidence in the revised manuscript to support the arguments in our work.

1. This work used PET membrane as a substrate for MOF preparation. This manuscript provides the thickness (10 μm) and pore diameter (220 nm) of the PET substrate. What is the channel density of the PET membranes? Is it a commercial PET membrane or prepared using a chemical etching method? If the 220 nm pore diameter is a statistical result, the author should provide the distribution and error bar of this data. It would be great if the effects of the pore size and pore density of the substrate on the quality of the MOF membranes could be studied and analyzed in this work.

Response: We thank this reviewer for the suggestion.

Commercial PET membranes with a channel density of 3×10^8 pore cm^{-2} have been used in our work. The statistical result of the pore diameter of the PET membrane is shown in Figure R1 (also, Figure S1c in the revised Supporting information). The average pore diameter of PET membrane was calculated as 227 ± 16.7 nm. The related data has been added into the revised manuscript and supporting information, and the sentences are as follows:

“Polyethylene terephthalate (PET) membrane with arrayed nanochannels (thickness, 12 μm ; pore diameter, 227 ± 16.7 nm) was used as the substrate for MOF preparation.” (Page 4, revised manuscript)

“PET membrane (12 μm in thickness; 3×10^8 pores cm^{-2}) was provided by Wuwei Kejin Xinfu Technology Co., Ltd.” (Page 15, revised manuscript)

We agree that the pore size and density might have impacts on the quality of the

prepared MOF membranes. However, these factors were not specifically examined in this work. This is first due to the use of commercial membranes, which poses challenges in controlling and regulating the pore size and density. In addition, we aim to study the impacts of surface group and pore size (< 1 nm) of MOF channels on the ion transport properties in this work. Therefore, a PET substrate with large pores (227 ± 16.7 nm) will not only ensure a fully grown of MOFs inside the pores, but also have negligible effect on the ion transport process in the MOF channels.

Figure R1. Pore diameter distribution of a commercial PET membrane. The average pore diameter is calculated as 227 ± 16.7 nm.

2. From Figure S1a, many pores coalesced together into larger pores. Does it affect the performance of MOF membranes? The authors should add more details of the PET membrane they used in this work. In addition, why does Figure S1b measure the thickness of the PET membrane as $12\ \mu\text{m}$, not $10\ \mu\text{m}$? Which data is correct?

Response: The coalescence of PET pores will not affect the performance of MOF membrane. As seen from the SEM images in Figure S2-S5, all the pores in the PET substrate, whether coalesced together or not, are filled with MOF materials. In addition, considering that the size of MOF channel (<1 nm) is much smaller than that of PET pore (>200 nm), ion transport through the membrane will dominantly affected by the MOF channel.

According to the suggestion of the reviewer, more details of the PET membrane have

been added in our revised manuscript, including the channel density (3×10^8 pore cm^{-2}), the pore diameter distribution (Figure R1), and the average pore diameter (227 ± 16.7 nm). The thickness of the PET membrane has also been further checked, and it is 12 μm . We have corrected this in the revised manuscript.

3. The authors claimed that monovalent/divalent cation selectivity of the UiO-66-(X)₂ membranes (Figure 3e) follows the order: UiO-66-(NH₂)₂ < UiO-66-(SH)₂ < UiO-66-(OH)₂ membrane. It would be better to provide the pore size distribution of the four tested MOF crystals to help understand the relationship between MOF pore sizes and membrane selectivity. In addition, it would be better to explain whether the ion selectivity observed in this work mainly depends on the membrane quality other than the MOF pore sizes because -SH group is larger than -OH group, and UiO-66-(SH)₂ should have better ion selectivity than UiO-66-(OH)₂ pores.

Response: According to the suggestion of this reviewer, the pore size distributions of UiO-66-(X)₂ have been obtained by using N₂ adsorption and desorption methods. As shown in Figure R2 (also, Figure S9 in the revised Supporting Information), there is hardly any difference between the pore size of UiO-66-(NH₂)₂, UiO-66-(SH)₂ and UiO-66-(OH)₂, which rules out the influence of pore size on the membrane selectivity in this experiment. We have added the results into the revised manuscript, and the sentences are as follows:

“The porosity of UiO-66, UiO-66-(NH₂)₂, UiO-66-(SH)₂ and UiO-66-(OH)₂ was examined by nitrogen adsorption analysis at 77 K, and their Brunauer-Emmett-Teller surface areas were 1060.9, 587.9, 559.0 and 715.4 m² g⁻¹, respectively (Figure S9). The pore size distribution curve derived from the N₂ adsorption isotherm indicates that the pore size of UiO-66-(X)₂ (X=NH₂, SH, and OH) is approximately 5.9 Å, exhibiting a similar size distribution.” (Page 5, revised manuscript)

In addition, the SEM and XRD characterization results demonstrate that dense and crack-free UiO-66-(X)₂ membranes with good crystallinities were obtained in this work (Figure 2a-e in the revised manuscript and Figure S2-5 in the revised Supporting Information), which rules out the influence of membrane quality on the membrane

selectivity in this experiment.

It is true that if only the size effect is considered, UiO-66-(SH)₂ might have better ion selectivity than UiO-66-(OH)₂ due to the slightly larger size of -SH group than -OH group. However, our results reveal that UiO-66-(OH)₂ exhibits a much better ion selectivity as compared to UiO-66-(SH)₂, which exactly shows the dominant influence of ion binding affinity on membrane selectivity in this experiment.

Figure R2. N₂ adsorption-desorption isotherms (a) and the pore size distributions (b) of UiO-66 and UiO-66-(X)₂ (X=NH₂, SH, OH) powders.

4. What is the diameter of the prepared MOF membrane? And what is the size for ion permselective testing?

Response: The area of the MOF membrane prepared in our work is approximately 120 mm² (diameter, 12 mm), and the size of the membrane for ion permselective testing is 3.14 mm² (diameter, 2 mm). We have added these data into the revised manuscript, and the sentences are as follows:

“A PET membrane (diameter, 12 mm) was placed vertically into a 25 mL reaction kettle.”
(Page 15, revised manuscript)

“The test area is approximately 3.14 mm².” (Page 18, revised manuscript)

5. The contact angles of UiO-66-(X)₂ membranes were measured (Figure S7) and all the membranes were slightly hydrophobic. From the SEM images of the UiO-66-(X)₂ membranes (Figure 2), the surfaces of the membranes were fully covered by MOF crystals. Thus, do you mean the UiO-66-(X)₂ are all hydrophobic?

Response: Thank this reviewer for pointing this out. Actually, all the membranes were first wetted with ethanol, which ensures the access of ions into the membrane in the following ion transport experiments. As revealed in Figure R3 (also, Figure S8 in the revised Supporting Information), the water contact angle of UiO-66-(OH)₂ membrane decreased from 127.9° to 103.4° after ethanol wetting, and it further decreased to 82.0° in 5 min. In order to avoid misleading, we have added the results in the revised manuscript and modified the related descriptions. The sentences are as follows:

“Therefore, all the membranes were first wetted with ethanol to ensure the access of ions into the membrane in the following ion transport experiments. As revealed in Figure S8, the water contact angle of UiO-66-(OH)₂ membrane decreased from 127.9° to 103.4° after ethanol wetting, and it further decreased to 82.0° in 5 min (Figure S8).”
(Page 5, revised manuscript)

Figure R3. Contact angles of the UiO-66-(OH)₂ membrane before and after ethanol wetting. (a) Contact angle of the as-prepared membrane. (b) Contact angle of the membrane after ethanol wetting. (c) Contact angle of (b) after 5 min.

6. The difference in mono-/di-valent ion selectivity was attributed to the inferior binding affinities (Page 6). Why could a stronger binding energy result in different ion flux trends for monovalent cations and divalent cations? A much clearer description is

required here.

Response: The difference in ion flux trends between monovalent and divalent ions is mainly attributed to the different rate-determining steps involved in their transmembrane transport. The ion flux of monovalent ions increases with the binding energy, while the ion flux of divalent ions decreases with the binding energy.

Specifically, the transmembrane transport of ions is mainly influenced by two distinct steps: the “entry” step and the “transportation” step. For the “entry” step, the binding of ions to the surface groups on the channel can help grab and stabilize the dehydrated ions. Therefore, a stronger binding energy will result in a lower energy barrier, that is, a higher ion flux. While in the “transportation” step, the binding of ions to the surface groups impedes their transport to the other side. Therefore, a stronger binding energy will result in a higher energy barrier, that is, a lower ion flux.

The rate-determining step may vary for different ions. As shown in Figure 3d, the conductance of monovalent ions gradually increases as the binding energy increases (that is, the functional group changes from -NH₂ to -SH to -OH). This indicates that the transport of monovalent ions is determined by the “entry” step. On the contrary, the conductance of divalent ions decreases as the binding energy increases, implying that their transport is determined by the “transportation” step. This difference can be attributed to the much inferior binding affinities of monovalent cations to the functional group as compared with divalent cations.

Therefore, it can be inferred that modifying the functional group to enhance the ion affinity can effectively enlarge the difference in flux between monovalent and divalent ions, resulting in a high ion selectivity.

7. On page 7, the authors said, “the conductance of monovalent cation in the UiO-66-(OH)₂ membrane is nearly equivalent to that in the bare PET membrane”. Data and figures should be given to prove this.

Response: The relevant data has been added to the revised Supporting Information (Figure R4, also Figure S13 in the revised Supporting Information).

Figure R4. Conductance of monovalent ions in UiO-66-(OH)₂ and bare PET membranes.

8. The purple line in Figure S12 is meaningless. It's not a curve to exhibit the data from the same samples. The purple line in Figure 4d is meaningless as well.

Response: We thank this reviewer for the suggestion. As shown in Figure R5-R6, in order to avoid misleading, the purple lines in these two figures have been removed in the revised manuscript (Figure 4d) and Supporting Information (Figure S15).

Figure R5. K⁺/Mg²⁺ ion selectivity at different activation energy differences (ΔE_a) (that is, ΔE_a of UiO-66 and UiO-66-(X)₂ (X=NH₂, SH, OH) membranes) in 100 mM electrolyte solutions. Here, ΔE_a=E_a(MgCl₂)-E_a(KCl).

Figure R6. Transport activation energy differences (ΔE_a) between K⁺ and Mg²⁺ ions and their relationship with K⁺/Mg²⁺ selectivity.

9. From Figure S14, the mono-/di-valent cation selectivity of the membranes increases with the increase of electrolyte concentration. Will it keep increasing when the electrolyte concentration is higher than 1 M?

Response: According to the suggestions of the reviewer, we measured the K⁺/Mg²⁺ selectivity over a broader range of electrolyte concentrations. As depicted in Figure R7 (also, Figure S17e in the revised Supporting Information), with the electrolyte concentration exceeding 1 M, the K⁺/Mg²⁺ selectivity reaches a plateau. This is because the transport of Mg²⁺ ions in the high concentration region gradually deviates from surface control, resulting in a gradual increase of the ion flux. This result has been added into the revised manuscript and Supporting Information, and the related descriptions are as follows:

“With the electrolyte concentration exceeding 1 M, the K⁺/Mg²⁺ selectivity reaches a plateau because the transport of Mg²⁺ ions in the high concentration region gradually deviates from surface control (Figure S17e).” (Page 9, revised manuscript)

Figure R7. K⁺/Mg²⁺ ion selectivity of UiO-66-(OH)₂ membrane at various concentrations (10⁻⁴-3 M).

10. Figure S17 exhibited the stability of UiO-66-(OH)₂ membranes within 48 h. How did the authors do this experiment? Was the membrane kept working under an electric field for 48 hours or just kept in water? More details should be provided to clarify what the stability is about. In addition, 48 h is not enough to exhibit the stability of the membrane.

Response: We thank this reviewer for the suggestions.

During the stability experiment, the membrane was sandwiched between two half electrolytic cells. We measured the ion selectivity every 24 h, and the membrane was stored in water between each test. To better evaluate the stability of the membrane, a much longer test was performed. As shown in Figure R8 (also, Figure S19 in the revised Supporting Information), the K⁺/Mg²⁺ selectivity of the membrane remained above 400 after 120 h. We have added these results into the revised manuscript, and the sentences are as follows:

“It is found that the K⁺/Mg²⁺ ion selectivity of the membrane remains basically unchanged within 120 h (Figure S19)” (Page 10, revised manuscript)

Figure R8. Stability of ion separation performance of UiO-66-(OH)₂ membrane. (a) K⁺ and Mg²⁺ ion conductance at different time. (b) K⁺/Mg²⁺ selectivity at different time.

11. UiO-66-(OMe)₂ membrane was obtained by post modification of UiO-66-(OH)₂ membrane. Is it only modified on the surface of UiO-66-(OH)₂ or fully modified even the MOFs into the PET channel? How do the authors identify the modification ratio of UiO-66-(OH)₂ MOFs?

Response: Thank this reviewer for the suggestion. The molecular size of CH₃I is calculated to be less than 2.6 Å, which is much smaller than the pore size of UiO-66-(OH)₂ (5.9 Å). Therefore, CH₃I molecule can easily get into the membrane and react with the -OH group. The modification ratio was determined by measuring the carbon content in UiO-66-(OH)₂ and UiO-66-(OMe)₂ using an elemental analyzer (Unicube, Germany). The mass percentages of carbon in UiO-66-(OH)₂ and UiO-66-(OMe)₂ were estimated as 27.27% and 28.17%, respectively, indicating that the modification ratio was 20%. Although the UiO-66-(OH)₂ membrane was not fully modified, the ion selectivity of the resulted membrane was significantly increased.

12. To evaluate the ion permselectivity of UiO-66-(OMe)₂ membrane, a mixed solution of 0.5 M KCl, NaCl, LiCl, MgCl₂, and CaCl₂ was prepared for testing. Why use this concentration in this experiment? In the other experiments, the authors used 0.1 M or 1 M for testing.

Response: There are two reasons for choosing a high electrolyte concentration in the permeation test: first, it guarantees a high ion selectivity; secondly, there is an increased chance of measuring the flux of divalent ions. In addition, to avoid precipitation in the

mixed solution with a high concentration (1 M), a concentration of 0.5 M was finally chosen in the experiment. We successfully demonstrated the ability of the MOF membrane in dealing with mixed electrolytes (Figure 4e and f). Although the electrolyte concentration might have a certain impact on the measured selectivity, it will not alter this conclusion.

13. It would be better to provide the MOF membrane mechanical properties for demonstrating their potential for separation application.

Response: The mechanical properties of the membrane indeed play a crucial role in practical membrane separation. They are determined by both the MOF materials and the porous substrate. In this work, we aim to explore the impacts of functional group and size of the sub-nanochannel on ion transport and ion separation. Therefore, the mechanical properties of the membrane are not within the scope of our discussion. The mechanical properties of the membrane are a topic that cannot be sidestepped when the membrane is used in practical applications. We will certainly pay more attention to the mechanical properties of the membrane in our future study.

Once again, we sincerely appreciate the valuable suggestions and comments provided by this reviewer.

Reviewer #2

In this work the authors prepared various functional metal organic framework (MOF) membranes (UiO-66-(X)₂, X=NH₂, SH, OH and OCH₃) and investigated the ionic transportation behavior through these membranes under electric field. The resulted membranes demonstrated good separation performance for the separation of monovalent and divalent cations due to the functional groups (X) and size of the MOF sub-nanochannel synergistically regulate the ion binding affinity and dehydration process. This may provide some clue for the development of high-precision ion separation membranes.

Response: We thank this reviewer for the valuable suggestions and comments. They

really help enhance the quality of our work.

1. The authors claimed they were inspired by the biological systems and designed the resulted membrane. The relationship between them is no so close. It is better to delete it.

Response: In this work, we carefully studied the principle of ion separation in biological channels and found that “ion-channel” interaction and channel size synergistically determine their exceptional performance. Drawing inspiration from this, we fabricated a series of MOF membranes (UiO-66-(X)₂, X=NH₂, SH, OH, and OCH₃), and systematically studied the impact of “ion-channel” interaction and channel size on monovalent/divalent ion separation.

Specifically, the cavity of K⁺ ion channel in a biological system can perfectly hold a dehydrated K⁺ ion, which indicates the important role of channel size. Furthermore, the channel can keep K⁺ ion stable via K⁺-carbonyl coordination. Up to four K⁺ ions are arranged inside the channel in close proximity, striking a balance between “ion-ion” repulsion and “ion-channel” attraction and leading to a low transport activation energy. This demonstrates the important role of “ion-channel” interaction. Therefore, a channel that matches the size of the target ion and has appropriate interaction with the target ion plays a decisive role in achieving efficient ion separation. This inspires us a lot in designing the MOF membranes in this work.

2. The synthesis process is simple. Both MOFs powders and crystals were form into the channel and on the surface of PET substrates, and in the solution. How to control the formation process only on the surface? How to guarantee the uniformity of MOFs on the surface and in the channels? Namely, why can MOFs crystals grow uniformly into the channels and on the surface? How about the effect of the channel size and surface property on the growth process?

Response: We thank this reviewer for raising these comments. There are three reasons to guarantee the uniformity of MOFs grown on the surface and in the channels. First, the carboxyl groups on the PET membrane surface and inner channel wall can easily

coordinate with Zr^{2+} ions, which promotes MOF growth. Second, the pore size of the PET channel exceeds 200 nm, which enables the easy access for Zr^{2+} ions and ligands, thus facilitating the simultaneous growth of MOF within the channels and on the surface. Third, an extended reaction time was adopted to ensure the sufficient growth of MOFs and improve the uniformity. Membranes with MOFs solely on the surface possess simpler structure, while they usually need special fabrication methods, such as interfacial growth method.¹⁻²

The pore size and surface properties indeed can impact the MOF growth process. The pore size will change the diffusion of Zr^{2+} ions and ligands into the PET pores, while surface property governs the interaction between PET substrate and MOFs. However, these factors were not specifically examined in this work. This is first due to the use of commercial membranes, which poses challenges in controlling and regulating the pore size and surface property. In addition, we aim to study the impact of surface group and pore size (< 1 nm) of MOF channels on the ion transport properties. Therefore, a PET substrate with large pores (>200 nm) and surface carboxyl groups will not only ensure a sufficient growth of MOFs inside the pores, but also have negligible effect on the ion transport process in the MOF channels.

3. The conductivity of pure bulk solution of the electrolytes should be given.

Response: Thank this reviewer for the suggestion. After removing the membrane in the set-up, we measured the ionic conductance of bulk electrolyte solutions. As shown in Figure R9a, the conductance is much larger than that through the MOF membranes, and electrolytes with divalent cations exhibit larger conductance than those with monovalent cations. In addition, the conductivities of pure bulk electrolyte solutions were also measured by a conductivity meter, and similar results were obtained (Figure R9b).

Figure R9. (a) Conductance of different electrolytes obtained by I-V curves. (b) Conductivities of different electrolytes measured by a conductivity meter.

4. How to determine the distribution of the mixture linkers into the MOFs? doping or just a mixture MOFs?

Response: Unfortunately, we did not come up with any good methods to determine the distribution of the mixture linkers into the MOFs because it is rather difficult to label and locate these linkers inside the MOF membrane. Considering that the ligands (Terephthalic acid and 2,5-dihydroxy terephthalic acid) are highly similar and well mixed in the reaction solution, we think the doping structure is preferred during MOF growth. Such doping MOF structures have also been obtained in other reported works using the similar fabrication method as this work.³⁻⁷

5. From the SEM cross-section, it is hard to see the full filled nanowires. It is better to remove the PET substrate and carry out the SEM or TEM observation.

Response: Thank this reviewer for raising these comments. The PET substrate is difficult to remove due to its solvent resistance properties. However, we repeated the SEM characterization and obtained images that can clearly show the cross-section of the MOF membranes. As shown in Figure R10-13 (also, Figure S2-5 in the revised Supporting Information), part of the MOF nanowires can be clearly seen at the upper, middle, and bottom sides of UiO-66, UiO-66-(NH₂)₂, UiO-66-(SH)₂ and UiO-66-(OH)₂ membranes, indicating fully filled nanowires. We have added these results into the revised Supporting Information.

Figure R10. Cross-sectional view of UiO-66 membrane.

Figure R11. Cross-sectional view of UiO-66-(OH)₂ membrane.

Figure R12. Cross-sectional view of UiO-66-(SH)₂ membrane.

Figure R13. Cross-sectional view of UiO-66-(NH₂)₂ membrane.

6. Why most of the I-V curves are not linear in the -1V to 1V?

Response: We appreciate this reviewer for bringing up this issue. We also noticed the phenomenon of non-linear I-V curves we obtained. As the ion affinity of the groups decreases from -OH to -SH to -NH₂, the I-V curves of K⁺, Na⁺, and Li⁺ electrolytes become more obviously non-linear (Figure 3a-c). Specifically, in the low voltage region,

the ion conductance is maintained at a small value, while in the high voltage region, the ion conductance increases rapidly. The reason for this phenomenon is that ions cannot overcome the transport energy barrier through the MOF sub-nanochannels at low driving voltages, but can overcome it at high driving voltages.

Specifically, the ion transport energy barrier is mainly related to two distinct steps: the “entry” step and the “transportation” step. For the “entry” step, the binding of ions to the surface groups on the channel can help grab and stabilize the dehydrated ions. Therefore, a stronger binding energy will result in a lower energy barrier. While for the “transportation” step, the binding of ions to the surface groups impedes their transport to the other side. Therefore, a stronger binding energy will result in a higher energy barrier.

Furthermore, as inferred from the result in Figure 3d, the transport energy barrier of monovalent ions mainly originates from the “entry” step because their conductance increases as the binding energy increases. On the contrary, the transport energy barrier of divalent ions mainly originates from the “transportation” step because their conductance decreases as the binding energy increases. Therefore, as the functional group changes from -OH to -SH to -NH₂, the transport energy barriers of K⁺, Na⁺, and Li⁺ ions become larger and larger, and their I-V curves become more and more non-linear. While for divalent ions, with the increase of affinity, that is, from -NH₂ to -SH to -OH, their transport energy barriers increase rapidly, resulting in low conductance in the entire voltage range.

This distinctive ion transport behavior observed in sub-nanochannels is of great research value, and can be applied to the construction of high-performance iontronics, as well as for biomimetic studies such as neural imitation. Our research group is currently conducting further investigations on this topic.

7. It is better to determine the activation energy by verifying temperature. This is more directly.

Response: Actually, the activation energy in this work was determined by verifying temperature. In order to present it clearer, we added the experimental details into the revised manuscript, and the sentences are as follows:

“The experiment process is as follows: first, the membrane was sandwiched between two temperature-controlled half cells; second, the I-V curve was recorded using an electrochemical workstation (CHI 660E) to obtain the corresponding ion conductance at each temperature when the set-up reached a steady state; third, the activation energy was calculated according to the formula by plugging the obtained conductance values. The temperature range used in this work was from 22 to 37 °C.” (Page 18, revised manuscript)

8. In Figure S17, Why the conductance for both KCl and CaCl₂ increase with time but selectivity decreases with time. Some leaking exists?

Response: We thank this reviewer for the comment. First, the conductance of KCl and MgCl₂ increases with time because the concentrations of K⁺ and Mg²⁺ ions inside the MOF membrane gradually increase due to their interactions with the surface group (Figure R14a; also, Figure S19a in the revised Supporting Information). Second, the increase of the conductance of MgCl₂ is slightly faster as compared with KCl, which might be attributed to the greater interaction between Mg²⁺ ion and MOF channel. This endows a gradual decrease of the K⁺/Mg²⁺ selectivity (Figure R14b; also, Figure S19b in the revised Supporting Information). We further checked the SEM images and XRD patterns of the membrane after the stability test. As shown in Figure R15 (also, Figure S20 in the revised Supporting Information), the integrity and crystallinity of the membrane were well maintained. Therefore, it can be inferred that there is no leakage during the test. We have added these results into the revised manuscript, and the sentences are as follows:

“It is found that the K⁺/Mg²⁺ ion selectivity of the membrane remains basically unchanged within 120 h (Figure S21), and the integrity and crystallinity of the

membrane were well maintained after the stability test (Figure S22), which demonstrates the high stability of the membrane.” (Page 10, revised manuscript)

Figure R14. Stability of ion separation performance of UiO-66-(OH)₂ membrane. (a) K⁺ and Mg²⁺ ion conductance at different time. (b) K⁺/Mg²⁺ selectivity at different time.

Figure R15. SEM images and XRD patterns of UiO-66-(OH)₂ membrane after the stability test. (a) Initial surface morphology of the membrane. (b) Surface morphology of the membrane after 120 h stability test. (c) XRD patterns of the membrane before and after stability test.

9. Since the membrane are composed of nanocrystals, many crystal grain boundaries should exist. Please address the contribution of these boundary to the ionic transportation.

Response: It is indeed crucial to recognize that both ion transport within MOF channels and along the crystal grain boundaries contribute to the total ion flux. However, as far as we know, there is currently a lack of effective methods to investigate and quantify the contributions of these two ion transport processes. One feasible approach is to use photoactive ions in combination with high spatial resolution spectroscopy, which might

enable the in-situ capture of ions within the MOF channels and at the grain boundaries. The behavior and mechanism of ion transport along grain boundaries will undoubtedly become one of our next research priorities.

10. From the water contact angles, the surfaces of the membranes are very hydrophobic. How can water fill into the nanochannels under the conditions as authors claimed.

Response: We thank this reviewer for pointing this out. Actually, all the membranes were first wetted with ethanol to ensure the access of ions into the membrane in the following ion transport experiment. As revealed in Figure R16 (also, Figure S8 in the revised Supporting Information), the water contact angle of UiO-66-(OH)₂ membrane decreased from 127.9° to 103.4° with ethanol wetting, and it further decreased to 82.0° in 5 min. In order to avoid misleading, we have added this result in the revised manuscript and modified the related descriptions. The sentences are as follows:

“Therefore, all the membranes were first wetted with ethanol to ensure the access of ions into the membrane in the following ion transport experiment. As revealed in Figure S8, the water contact angle of UiO-66-(OH)₂ membrane decreased from 127.9° to 103.4° after ethanol wetting, and it further decreased to 82.0° in 5 min (Figure S8).” (Page 5, revised manuscript)

Figure R16. Contact angles of the UiO-66-(OH)₂ membrane before and after ethanol wetting. (a) Contact angle of the as-prepared membrane. (b) Contact angle of the membrane after ethanol wetting. (c) Contact angle of (b) after 5 min.

11. Is it possible to perform the membrane for pressure-driven ionic separation? Pressure-driven ionic separation is more facile.

Response: We thank this reviewer for the valuable suggestion. The pressure-driven method is more convenient, but it requires specific mechanical properties for the membranes. In this work, we mainly focus on the exploration of ion transport mechanism inside the MOF sub-nanochannels. Therefore, the simple permeation method is adopted in the mixed-electrolyte separation experiment. In future work, we will consider pressure-driven ionic separation and push our separation membranes to practical applications.

Reviewer #3

This manuscript titled in “Regulating Ion Affinity and Dehydration of Metal-Organic Framework Sub-nanochannels for High-Precision Ion Separation” by Xia et al. reports the UiO-66-(OCH₃)₂ membrane to achieve the high K⁺/Mg²⁺ selectivity as 1567.8. This work provides a new gateway to the understanding of ion transport mechanism and development of high-precision ion separation membranes. I would recommend the acceptance of the manuscript for a major revision after addressing the following issues.

Response: We thank this reviewer for the valuable suggestions and comments. They really help us to enhance the quality of our work.

1. Why do not the I-V curves in Figure 3c follow the Ohm’s law?

Response: We appreciate the reviewer for bringing up this issue. We also noticed the phenomenon of non-linear I-V curve during our experiments. As the ion affinity of the groups decreases from -OH to -SH to -NH₂, the I-V curves of K⁺, Na⁺, and Li⁺ electrolytes become more obviously non-linear (Figure 3a-c). Specifically, in the low voltage region, the ion conductance is maintained at a small value, while in the high voltage region, the ion conductance increases rapidly. The reason for this phenomenon is that ions cannot overcome the transport energy barrier through the MOF sub-nanochannels at low driving voltages, but can overcome it at high driving voltages.

Specifically, the ion transport energy barrier is mainly related to two distinct steps: the “entry” step and the “transportation” step. For the “entry” step, the binding of ions to the surface groups on the channel can help grab and stabilize the dehydrated ions. Therefore, a stronger binding energy will result in a lower energy barrier. While for the “transportation” step, the binding of ions to the surface groups impedes their transport to the other side. Therefore, a stronger binding energy will result in a higher energy barrier.

Furthermore, as inferred from the result in Figure 3d, the transport energy barrier of monovalent ions mainly originates from the “entry” step because their conductance increases as the binding energy increases. On the contrary, the transport energy barrier of divalent ions mainly originates from the “transportation” step because their conductance decreases as the binding energy increases. Therefore, as the functional group changes from -OH to -SH to -NH₂, the transport energy barriers of K⁺, Na⁺, and Li⁺ ions become larger and larger, and their I-V curves become more and more non-linear. While for divalent ions, with the increase of affinity, that is, from -NH₂ to -SH to -OH, their transport energy barrier increases rapidly, resulting in low conductance in the entire voltage range.

This distinctive ion transport behavior observed in sub-nanochannels is of great research value, and can be applied to the construction of high-performance iontronics, as well as for biomimetic studies such as neural imitation. Our research group is currently conducting further investigations on this topic.

2. “When the electrolyte concentration is increased from 10⁻⁴ to 1 M, the conductance of K⁺ ion linearly increases, while the conductance of Mg²⁺ ion keeps almost constant (Figure S14).” Why can it show excellent surface charge control at a high concentration of 1 M? Referring to other literatures, the electric double layer will become very thin at high concentrations.

Response: We thank this reviewer for the suggestion. We repeated this experiment and adopted a broader MgCl₂ concentration range (10⁻⁴-3 M). As shown in Figure R17 (also,

Figure S17d in the revised Supporting Information), the conductance fluctuates when the concentration is lower than 1 M, indicating a surface-controlled ion transport. While, the conductance gradually increases when the concentration is higher than 1 M, indicating the deviation of ion transport from surface control. This phenomenon is mainly attributed to two reasons. First, the size of the MOF channel (0.59 nm) is significantly smaller than that of a traditional nanochannel (>10 nm). The electric double layer has a thickness of 0.3 nm at 1 M concentration, which still occupies most of the MOF channel. Second, due to the divalent nature of Mg^{2+} , it exhibits a much stronger electrostatic interaction with the channel surface as compared to monovalent ions, thus resulting in a more pronounced surface-controlled transport behavior.

Figure R17. Transmembrane conductance of MgCl_2 at different concentrations (10^{-4} -3 M).

3. Why can the transport of monovalent cations be controlled by the bulk solution? Why cannot the transport of monovalent cations be controlled by the surface charge? Here needs more discussions.

Response: We measured the conductance of KCl again in a broader concentration range (10^{-7} -3 M). As shown in Figure R18 (also, Figure S17b in the revised Supporting Information), as the concentration decreases, the ion conductance linearly decreases, and then deviates from the bulk value when the concentration is lower than 10^{-4} M, showing surface-controlled transport behavior.

Figure R18. Transmembrane conductance of KCl at different concentrations (10^{-7} -3 M).

4. The stability test time (48 h) is too short to explain high stability of the membrane.

Response: We thank this reviewer for the suggestion. To better evaluate the stability of the membrane, a much longer test was performed. As shown in Figure R19 (also, Figure S19 in the revised Supporting Information), the ion selectivity of the membrane remained above 400 after 120 h. We have added this result into the revised manuscript, and the sentences are as follows:

“It is found that the K^+/Mg^{2+} ion selectivity of the membrane remains basically unchanged within 120 h (Figure S19)” (Page 10, revised manuscript)

Figure R19. Stability of ion separation performance of UiO-66-(OH)₂ membrane. (a) K^+ and Mg^{2+} ion conductance at different time. (b) K^+/Mg^{2+} selectivity at different time.

5. Why use UiO-66-(OH)₂ as the membrane rather than UiO-66-(COOH)₂? Here needs more discussions.

Response: Thank this reviewer for the suggestion. In this work, one of our main research objects is to investigate the influence of different groups with varying ion affinities on the performance of ion separation in MOFs. This can be realized by selecting -OH, -SH and -NH₂ as functional groups because they are similar in size but possess different ion affinities. However, if UiO-66-(COOH)₂ was chosen, we could not find a suitable UiO-66-type materials for comparison. Additionally, due to the smaller size of -OH, we were able to convert -OH into -OMe through post-modification. This allows us to further regulate the channel size and explore its impact on ion separation. As a result, UiO-66-(OH)₂ was ultimately chosen for this study.

Once again, we sincerely appreciate the valuable suggestions and comments provided by this reviewer.

Reference

1. Zhang, X. et al. Electrochemically assisted interfacial growth of MOF membranes. *Matter* **1**, 1285-1292 (2019).
2. Zhang, Y. et al. Crystal seeds induced interfacial growth of zirconium metal-organic framework membranes towards efficient hydrogen purification. *J. Membr. Sci.* **689**, 122171 (2024).
3. Hou, Q. et al. Ultra-tuning of the aperture size in stiffened ZIF-8_Cm frameworks with mixed-linker strategy for enhanced CO₂/CH₄ separation. *Angew. Chem. Int. Ed.* **58**, 327-331 (2019).
4. Zhou, S. et al. Asymmetric pore windows in MOF membranes for natural gas valorization. *Nature* **606**, 706-712 (2022).
5. Deng, H. et al. Multiple functional groups of varying ratios in metal-organic frameworks. *Science* **327**, 846-850 (2010).
6. Hou, J., Zhang, H., Wang, H., Thornton, A.W. & Konstas, K. Amphoteric metal-

organic framework subnanochannels with pH-tuneable cation and anion sieving properties. *J. Mater. Chem. A* **11**, 13223-13230 (2023).

7. Truong, B.N. et al. Tuning hydrophilicity of aluminum MOFs by a mixed-linker strategy for enhanced performance in water adsorption-driven heat allocation application. *Adv. Sci.* **10**, 2301311 (2023).

REVIEWER COMMENTS

Reviewer #1 (Remarks to the Author):

The authors have addressed most of the comments carefully, and the manuscript looks much better. However, the comments on membrane quality and MOF pore sizes and the relationships between membrane selectivity and MOF pore structures have not been well investigated.

1. In Figure S9, the authors tested the BET surface area of the four MOF crystal powders.

Strangely, however, pore volumes of small and large cavities of the UiO-66-(SH)₂ and UiO-66-(OH)₂ could not be seen from the graphs. How do the pore sizes change when modifying UiO-66-(OH)₂ to UiO-66-(OMe)₂?

2. For the XRD patterns of MOF-PET membranes in Figure 2e, why are the peaks of UiO-66-(OH)₂ and UiO-66-(NH₂)₂ not obvious compared to the UiO-66. It would be better to provide XRD patterns of MOF powders synthesized under the same conditions to confirm their crystal structures.

3. For MD simulations, it is excellent that energy barriers of K⁺ and Mg²⁺ through UiO-66-(OH)₂ and UiO-66-(OMe)₂ have been studied. As the authors claim that UiO-66-(OH)₂ has better ion selectivity than UiO-66-(SH)₂ and UiO-66-(NH₂)₂, it would be better to confirm these theoretically. For example, if the simulated results show UiO-66-(SH)₂ has higher energy barriers for Mg²⁺ than UiO-66-(OH)₂, the lower ion selectivity of UiO-66-(SH)₂ would be attributed to defects of the membrane, as illustrated by the XRD patterns.

4. How about the stability of the UiO-66-(OMe)₂? The XRD patterns of the UiO-66-(OMe)₂ powder and membranes before and after testing should be provided to illustrate its stability for ion separation.

Reviewer #2 (Remarks to the Author):

The comments have been well addressed. The current form is acceptable.

Reviewer #3 (Remarks to the Author):

First of all, we would like to thank the author for his earnest reply. There are still some questions to be answered.

1. The mechanism of ion transport behavior needs to be explained for Question 3.

2. The stability decreased significantly after 120 h. Why? The stability test time could be longer in future research.

RESPONSE TO REVIEWERS' COMMENTS

Reviewer #1

The authors have addressed most of the comments carefully, and the manuscript looks much better. However, the comments on membrane quality and MOF pore sizes and the relationships between membrane selectivity and MOF pore structures have not been well investigated.

Response: We thank this reviewer again for the valuable comments on our revised manuscript. We have carefully studied them and made a further revision to improve the quality of our work.

1. In Figure S9, the authors tested the BET surface area of the four MOF crystal powders. Strangely, however, pore volumes of small and large cavities of the UiO-66-(SH)₂ and UiO-66-(OH)₂ could not be seen from the graphs. How do the pore sizes change when modifying UiO-66-(OH)₂ to UiO-66-(OMe)₂?

Response: We thank this reviewer for the comment. In order to better present the results, we optimized the sample preparation and testing methods, and then performed the nitrogen adsorption analysis again. As shown in Figure R1 (also, Figure S10 in the revised Supporting Information), the pore volumes of small and large cavities of the UiO-66-(SH)₂ and UiO-66-(OH)₂ can be clearly distinguished. In addition, the pore size distribution of UiO-66-(OMe)₂ was also measured and shown in Figure R2 (also, Figure S26 in the revised Supporting Information). It is revealed that the pore size of UiO-66-(OMe)₂ is approximately 5.0 Å, which is slightly smaller as compared to that of UiO-66-(OH)₂. We have added these results into the revised manuscript as follows:

“The porosity of UiO-66-(OMe)₂ was examined by nitrogen adsorption analysis at 77 K, and the Brunauer-Emmett-Teller surface area was 583.9 m² g⁻¹ (Figure S26). The pore size distribution curve derived from the N₂ adsorption isotherm indicates that the pore size of UiO-66-(OMe)₂ is approximately 5.0 Å, which is slightly smaller as compared to that of UiO-66-(OH)₂.” (Page 10, revised manuscript)

Figure R1. Nitrogen adsorption-desorption isotherm (a) and pore size (b) of UiO-66 and UiO-66-(X)₂ (X=OH, SH, NH₂).

Figure R2. Nitrogen adsorption-desorption isotherm (a) and pore size (b) of UiO-66-(OMe)₂.

2. For the XRD patterns of MOF-PET membranes in Figure 2e, why are the peaks of UiO-66-(OH)₂ and UiO-66-(NH₂)₂ not obvious compared to the UiO-66. It would be better to provide XRD patterns of MOF powders synthesized under the same conditions to confirm their crystal structures.

Response: We thank this reviewer for the suggestion. In order to provide more precise results, we repeated the XRD measurement with improved sample preparation and testing methodologies. As shown in the Figure R3 (also, Figure 2e in the revised manuscript), the peaks of UiO-66-(OH)₂ and UiO-66-(NH₂)₂ can be clearly distinguished. In addition, we also measured the XRD patterns of MOF powders synthesized under the same conditions. As shown in the Figure R4 (also, Figure S6 in the revised Supporting Information), the main peaks of the XRD pattern of UiO-66-(X)₂ (X=NH₂, SH, OH) membrane are located at the same diffraction angles as those of UiO-66-(X)₂ (X=NH₂, SH, OH) powders, confirming their crystal structures.

Figure R3. XRD patterns of PET, UiO-66, and UiO-66-(X)₂ (X=OH, SH, NH₂) membranes.

Figure R4. XRD patterns of UiO-66 and UiO-66-(X)₂ (X=OH, SH, NH₂) powders.

3. For MD simulations, it is excellent that energy barriers of K⁺ and Mg²⁺ through UiO-66-(OH)₂ and UiO-66-(OMe)₂ have been studied. As the authors claim that UiO-66-(OH)₂ has better ion selectivity than UiO-66-(SH)₂ and UiO-66-(NH₂)₂, it would be better to confirm these theoretically. For example, if the simulated results show UiO-66-(SH)₂ has higher energy barriers for Mg²⁺ than UiO-66-(OH)₂, the lower ion selectivity of UiO-66-(SH)₂ would be attributed to defects of the membrane, as illustrated by the XRD patterns.

Response: We thank this reviewer for the suggestion. Actually, DFT calculations instead of MD simulation have been performed in this work to reveal the binding energies between UiO-66-(X)₂ and ions (Table S1 in the Supporting Information), which can help to confirm that UiO-66-(OH)₂ has better ion selectivity than UiO-66-(SH)₂ and UiO-66-(NH₂)₂.

As seen from Table S1, as compared to the monovalent cations (Li⁺, Na⁺ and K⁺),

divalent cations (Mg^{2+} and Ca^{2+}) show much larger binding energies to UiO-66-(X)_2 . This indicates that divalent cations interact much more intensively with the functional groups due to their higher ionic charges, which will severely impede their transport. This is also confirmed by the higher transport activation energy observed for MgCl_2 as compared to that for KCl (experimental results, Figure S15 in Supporting Information). In addition, the calculated binding energy for Mg^{2+} ion follows the order: $\text{UiO-66-(NH}_2)_2 < \text{UiO-66-(SH)}_2 < \text{UiO-66-(OH)}_2$, revealing the strongest interaction between Mg^{2+} ion and UiO-66-(OH)_2 . This is also confirmed by the highest transport activation energy for MgCl_2 through UiO-66-(OH)_2 (experimental results, Figure S15 in Supporting Information). The ΔE_a value between K^+ and Mg^{2+} is further calculated (Figure S16 in Supporting Information), and it follows the order: $\text{UiO-66-(NH}_2)_2 < \text{UiO-66} < \text{UiO-66-(SH)}_2 < \text{UiO-66-(OH)}_2$. Since the ion selectivity exponentially increases with the increase of ΔE_a , UiO-66-(OH)_2 exhibits better ion selectivity than UiO-66-(SH)_2 and $\text{UiO-66-(NH}_2)_2$.

Therefore, combining the theoretical calculation and experimental results, we conclude that through regulating the functional group, the ion binding affinity can be effectively modulated to tune the ion transport activation energy to improve the ion selectivity.

4. How about the stability of the UiO-66-(OMe)_2 ? The XRD patterns of the UiO-66-(OMe)_2 powder and membranes before and after testing should be provided to illustrate its stability for ion separation.

Response: We thank this reviewer for the suggestion. As seen from Figure R5 (also, Figure S29 in the revised Supporting Information), the $\text{K}^+/\text{Mg}^{2+}$ selectivity of the UiO-66-(OMe)_2 membrane remains larger than 1000 in the first 4 days (96 h) and then decreases to 928 after 5 days (120 h). The decrease of the ion selectivity is mainly attributed to the gradual intrusion of the Mg^{2+} ions into the membrane, which increases its conductance (Figure R5a). The SEM images and XRD patterns of the membrane after the stability test were further obtained. As shown in Figure R6 and Figure R7 (also,

Figure S30 and Figure S23b in the revised Supporting Information), the integrity and crystallinity of the membrane were well maintained. We have added these results into the revised manuscript as follows:

“It is also revealed that the K^+/Mg^{2+} selectivity and the structure of the membrane can be well maintained within 120 h (Figure S29-S30).” (Page 11, revised manuscript)

Figure R5. Stability of ion separation performance of UiO-66-(OMe)₂ membrane. (a) Relationship between ion conductance and time. (b) Relationship between ion selectivity and time.

Figure R6. SEM images and XRD patterns of UiO-66-(OMe)₂ membrane after the stability test. (a) Initial surface morphology of the membrane. (b) Surface morphology of the membrane after 120 h stability test. (c) XRD patterns of the membrane before and after stability test.

Figure R7. XRD pattern of UiO-66-(OMe)₂ powder.

Reviewer #2

The comments have been well addressed. The current form is acceptable.

Response: We are very grateful to the reviewer for the help in improving the quality of this article.

Reviewer #3

First of all, we would like to thank the author for his earnest reply. There are still some questions to be answered.

Response: Thank this reviewer for the valuable comments on our revised manuscript. The questions have been carefully studied and addressed.

1. The mechanism of ion transport behavior needs to be explained for Question 3.

Response: We thank this reviewer for the suggestion. As compared to the divalent Mg²⁺ ion, the monovalent K⁺ ion has a much weaker electrostatic interaction with the channel

surface, thus resulting in a more unobvious surface-controlled transport behavior. As seen from Figure R8, the transport of K^+ ion is controlled by the bulk solution in the concentration range of 10^{-4} to 3 M. However, when the concentration is lower than 10^{-4} M, the conductance of K^+ ion deviates from the bulk value. This is because that the charge shielding effect becomes weaker under the low ionic strength condition, and the electrostatic interaction between ions and surface charge starts to dominate the ion transport in the narrowed MOF channel (0.59 nm).

Figure R8. Transmembrane conductance of KCl at different concentrations (10^{-7} -3 M).

2. The stability decreased significantly after 120 h. Why? The stability test time could be longer in future research.

Response: We thank this reviewer for the comment. As seen from Figure R9a (also, Figure S20a in the revised Supporting Information), the conductance of both KCl and $MgCl_2$ increases with time because the concentrations of K^+ and Mg^{2+} ions inside the MOF membrane gradually increase due to their interactions with the surface group. In addition, the increase of the conductance of $MgCl_2$ is slightly faster as compared to KCl. This can be attributed to the greater interaction between Mg^{2+} ion and MOF channel, which facilitates its intrusion into the membrane. This ends a gradual decrease of the K^+/Mg^{2+} selectivity (Figure R9b; also, Figure S20b in the revised Supporting Information). There is no doubt that stability is crucial in membrane separation applications. Therefore, in our future research, we will further optimize the channel

structure to secure long-term operational stability for membrane separation.

Figure R9. Stability of ion separation performance of UiO-66-(OH)₂ membrane. (a) K⁺ and Mg²⁺ ion conductance at different time. (b) K⁺/Mg²⁺ selectivity at different time.

REVIEWERS' COMMENTS

Reviewer #1 (Remarks to the Author):

The authors have carefully addressed all my concerns. I have no further comments. The manuscript is ready for publication.

Reviewer #3 (Remarks to the Author):

The reviewer suggests the acceptance of the revises manuscript.